# Decomposition of three aerosol types using lidar-derived depolarization ratios at two wavelengths

Xiaoxia Shang[1], Maria Filioglou[1], Julian Hofer[2], Moritz Haarig[2], Qiaoyun Hu[3], Philippe Goloub[3], Sami Romakkaniemi[1], and Mika Komppula[1]

[1]Atmospheric Research Centre of Eastern Finland, Finnish Meteorological Institute, Kuopio, Finland.
[2]Leibniz Institute for Tropospheric Research, Leipzig, Germany
[3]Univ. Lille, CNRS, UMR 8518 – LOA – Laboratoire d'Optique Atmosphérique, 59000 Lille, France

**Correspondence:** Xiaoxia Shang (xiaoxia.shang@fmi.fi)

**Abstract.** Lidar-based algorithms for aerosol-type separation have the potential to improve air-quality assessments, estimates of aerosol direct and indirect radiative forcing, and the detailed characterization of their vertical distribution. In this study, we present an easy-to-apply algorithm that employs lidar-derived particle linear depolarization ratios measured at two wavelengths to separate up to three aerosol-type-specific particle backscatter fractions. These fractions are estimated under the
assumptions that the depolarization ratios of each aerosol type in the mixture differ, and that both the depolarization ratios and the backscatter-related Ångström exponents at two wavelengths for each aerosol type are known. The mathematical relationship between particle linear depolarization ratios at two wavelengths for an aerosol mixture has been derived and expressed as a system of equations. These equations define the region of the observational space that can be meaningfully populated, with boundaries determined by the depolarization ratios and backscatter-related Ångström exponents of the pure aerosol types.
Data collected in the Arabian Peninsula confirmed the predicted region of the observational space. The proposed algorithm is applied to synthetic dust mixtures as well as to atmospheric lidar observations of Arabian dust, Asian dust, Saharan dust and their mixtures, with the goal of decomposing coarse-mode dust, fine-mode dust, and low-depolarizing non-dust aerosols. We also discuss the impact of uncertainties in the prior optical properties of the pure aerosol types, along with the effects of observational uncertainties and biases. Overall, the method enhances the potential of dual-wavelength depolarization measurements
for improving our understanding of the vertical distribution of coarse and fine dust.

## 1 Introduction

Light detection and ranging (lidar) is a powerful remote sensing technique that provides vertical information of atmospheric aerosol and clouds from ground and space at high vertical resolution. Irregularly shaped particles induce strong depolarization of laser light. Polarization-based algorithms have been developed to separate the aerosol profiles of weakly light depolarizing
(assumed near-spherical, e.g., anthropogenic haze, biomass burning smoke, maritime) and strongly light depolarizing (non-spherical, e.g., volcanic ash, desert dust, pollen) particles (e.g., Shimizu et al., 2004; Tesche et al., 2009, 2011; Ansmann et al., 2011, 2012; Sugimoto and Lee, 2006; Nishizawa et al., 2007; Freudenthaler et al., 2009; Groß et al., 2011, 2012; Miffre et al., 2012; Burton et al., 2014; Nisantzi et al., 2014; Shang et al., 2020). Such techniques have been primarily used for decoupling

the particle backscatter coefficient profiles of, e.g., dust and non-dust particles (Tesche et al., 2009), volcanic ash and fine-mode particles (Ansmann et al., 2011; Marenco and Hogan, 2011), and pollen particles and non-depolarizing background aerosol (Shang et al., 2022). In addition, the POLIPHON (POlarization LIdar PHOtometer Networking) method (Ansmann et al., 2012) allows the retrieval of the particle number, surface area, and volume/mass concentration of fine-mode and coarse-mode particles, in synergy with sun-sky photometer measurements (one-step POLIPHON). The extended POLIPHON method, namely the two-step POLIPHON method, further allows the separation of non-dust, fine dust, and coarse dust particle contributions of the above-mentioned optical and microphysical quantities (Mamouri and Ansmann, 2014, 2017). The two-step POLIPHON method comprises two subsequent phases similar to the one-step POLIPHON, and allows separating three aerosol types, as long as their particle depolarization ratios are distinct. However, to separate the backscatter coefficients of coarse dust in the first step, it requires an assumption about the spherical particle fraction to estimate the overall depolarization ratio for the residual aerosol (a mixture of non-dust and fine dust with unknown mixing ratio). The POLIPHON method has been applied to many lidar observations across the world, benefiting from the wide availability of these single-wavelength polarization lidars (Ansmann et al., 2019; Córdoba-Jabonero et al., 2018; Mamali et al., 2018; Haarig et al., 2019; Proestakis et al., 2024). David et al. (2013) presented the first study using a dual-wavelength polarization lidar combined with T-matrix simulations to partition a three-component external aerosol mixture. Their approach required lidar-derived co- and cross-polarized backscatter coefficients and relied heavily on modeled optical properties and assumed particle size distributions. While this methodology demonstrated the feasibility of separating complex mixtures, it involves significant computational effort and depends on prior knowledge of particle microphysical characteristics. Veselovskii et al. (2022) proposed a method that combines the particle depolarization ratio at 532 nm with fluorescence capability to distinguish smoke, dust, pollen, and urban aerosol particles. However, the application of fluorescence is limited by solar background noise, making observations feasible primarily at nighttime or in low-light conditions.

Most commonly, the 532 nm and 355 nm wavelengths have been used to perform lidar-derived depolarization ratio measurements. Most lidar stations in the European Aerosol Research Lidar Network (EARLINET, Pappalardo et al. (2014); https://www.earlinet.org, last access: 10 October 2024) and the Raman and polarization lidar network (PollyNET, Baars et al. (2016); https://polly.tropos.de, last access: 10 October 2024) measure the particle linear depolarization ratios at 532 and/or 355 nm. The NASA Micro-Pulse Lidar Network (MPLNET, Welton et al. (2001); https://mplnet.gsfc.nasa.gov, last access: 10 October 2024) provide polarization measurements at 532 nm. The space-borne lidars CALIOP (Cloud-Aerosol LIdar with Orthogonal Polarization) onboard the CALIPSO (the Cloud-Aerosol Lidar and Infrared Pathfinder Satellite Observation) (Winker et al., 2009), and ATLID (Atmospheric Lidar, Donovan et al., 2024) in the ESA's EarthCARE (Cloud, Aerosol and Radiation Explorer; https://earth.esa.int/eogateway/missions/earthcare, last access: 10 October 2024) mission retrieve depolarization ratio at 532 and 355 nm, respectively. Depolarization ratio observations at 1064 nm have been increasing the last years as well (Haarig et al., 2022; Hu et al., 2020). The Vaisala CL61 ceilometer, and the HALO Photonics Stream Line Pro scanning Doppler lidar (Pearson et al., 2009) also provide polarization observations at 910 nm and 1565 nm, respectively. The utilization of the multi-wavelength polarization lidar or a synergy of multiple lidars with the capability to provide concurrent depolarization ratio observations at various wavelengths allows the investigation of the spectral dependence on the depolarization

ratios. This is an important aspect to characterize different aerosol types, such as dust (Haarig et al., 2017a, 2022) and pollen
(Bohlmann et al., 2021; Filioglou et al., 2023). In particular, it could enable the distinction of non-spherical aerosol types.

In this study, we present a methodology that uses the particle linear depolarization ratios measured at two wavelengths to estimate the particle backscatter fractions of two (Sect. 2.1) or three (Sect. 2.2) aerosol types. This approach builds upon well-developed polarization-based algorithms, such as the POLIPHON method, without introducing assumptions of the spherical particle fractions. A unique mathematical solution for the backscatter fractions can be obtained, if the optical properties of each aerosol type are well characterized, and the particle linear depolarization ratio is measured with high accuracy. The particle linear depolarization ratio is a parameter widely used in the lidar community, which is also a standard output product from lidar networks. The relationship between particle linear depolarization ratios at two wavelengths are also investigated mathematically for mixtures of two and three aerosol types. The proposed easy-to-apply algorithm is first applied to synthetic dust mixtures (Sect. 2.3), followed by a comprehensive sensitivity analysis (Sect. 2.4). It is then applied to lidar observations of aerosols in different dust-affected regions (Sect. 3).

## 2 Methodology

The basic concept is to use lidar-derived particle linear depolarization ratios obtained from a multi-wavelength lidar (or two separate instruments), together with two key optical properties (namely, the particle depolarization ratios and the backscatter-related Ångström exponents for pure aerosol types), to separate the aerosol mixture into its individual aerosol types. In this section, we introduce the algorithm and the corresponding set of equations for decomposing mixtures of two or three aerosol types (Sects. 2.1–2.2). The algorithm is then applied to synthetic aerosol mixtures in Sect. 2.3, followed by a comprehensive sensitivity and uncertainty analysis in Sect. 2.4.

To begin with, the particle linear depolarization ratio of a particle ensemble at wavelength $\lambda$, denoted as $\delta_p(\lambda)$, can be expressed by Eqs. 1–2 for an aerosol mixture of two or three externally mixed aerosol types (Shimizu et al., 2004; Tesche et al., 2009; Mamouri and Ansmann, 2014). In this context, each particle consists of a single aerosol type, and the calculation involves the aerosol backscatter coefficient ($\beta_x$) and the particle depolarization ratio ($\delta_x$), where the index $x$ corresponds to each aerosol type ($a$, or $b$, or $c$), or to the mixture ($p$).

$$\delta_p(\lambda) = \frac{\frac{\beta_a(\lambda)\delta_a(\lambda)}{\delta_a(\lambda)+1} + \frac{\beta_b(\lambda)\delta_b(\lambda)}{\delta_b(\lambda)+1}}{\frac{\beta_a(\lambda)}{\delta_a(\lambda)+1} + \frac{\beta_b(\lambda)}{\delta_b(\lambda)+1}} \tag{1}$$

$$\delta_p(\lambda) = \frac{\frac{\beta_a(\lambda)\delta_a(\lambda)}{\delta_a(\lambda)+1} + \frac{\beta_b(\lambda)\delta_b(\lambda)}{\delta_b(\lambda)+1} + \frac{\beta_c(\lambda)\delta_c(\lambda)}{\delta_c(\lambda)+1}}{\frac{\beta_a(\lambda)}{\delta_a(\lambda)+1} + \frac{\beta_b(\lambda)}{\delta_b(\lambda)+1} + \frac{\beta_c(\lambda)}{\delta_c(\lambda)+1}} \tag{2}$$

Moreover, the aerosol backscatter fraction $\phi_x$ (Eq. 3) is defined as the ratio of the backscatter coefficient of the aerosol type $x$ ($\beta_x$) to the total particle backscatter coefficient ($\beta_p$). The sum of all aerosol backscatter fractions in the mixture is equal to 1.

$$\phi_x(\lambda) = \frac{\beta_x(\lambda)}{\beta_p(\lambda)} \tag{3}$$

The backscatter-related Ångström exponent ($\text{Å}_{\beta,x}$; Ångström, 1964) between two wavelengths, $\lambda_1$ and $\lambda_2$, can be expressed through Eq. 4 (the index $x$ can be used for one aerosol type or for the mixture), where parameter $\eta$, often refereed as the backscatter color ratio, is a function of $\text{Å}_\beta$ and it is defined as follows:

$$\eta_x = \left(\frac{\lambda_1}{\lambda_2}\right)^{-\text{Å}_{\beta,x}(\lambda_1,\lambda_2)} = \frac{\beta_x(\lambda_1)}{\beta_x(\lambda_2)}. \tag{4}$$

## 2.1 Mixture of two aerosol types

As a first step, a mixture of two aerosol types is considered assuming a strongly depolarizing type and a weakly depolarizing type, with type denoted by the subscripts $a$ and $b$. The particle depolarization ratios of the two aerosol types should be different (i.e., $\delta_a \neq \delta_b$). The total particle backscatter coefficient, $\beta_p$, is the sum of the backscatter coefficients of aerosols $a$ and $b$ ($\beta_p = \beta_a + \beta_b$). The separation method using $\delta_p$ measured at one wavelength, initially developed by Shimizu et al. (2004) and thoroughly discussed in Tesche et al. (2009), is well known and has been widely utilized. In this section, we present the approach in an alternative form, specifically to derive the mathematical relationship between lidar-derived particle depolarization ratios at two wavelengths ($\delta_p(\lambda_1)$ and $\delta_p(\lambda_2)$).

Using Eqs. 1 and 3, $\delta_p$ of the mixture at wavelength $\lambda_i$ ($i = 1, 2$) can be expressed by Eq. 5. Also, the sum of the two aerosol backscatter fractions at $\lambda_i$ is equal to 1 (Eq. 6).

$$\delta_p(\lambda_i) = \frac{\frac{\phi_a(\lambda_i)\delta_a(\lambda_i)}{\delta_a(\lambda_i)+1} + \frac{\phi_b(\lambda_i)\delta_b(\lambda_i)}{\delta_b(\lambda_i)+1}}{\frac{\phi_a(\lambda_i)}{\delta_a(\lambda_i)+1} + \frac{\phi_b(\lambda_i)}{\delta_b(\lambda_i)+1}} \tag{5}$$

$$\phi_a(\lambda_i) + \phi_b(\lambda_i) = 1 \tag{6}$$

For the sake of simplicity, an aerosol-type-dependent term, $Q_x(\lambda_i)$, has been introduced, which is based on the particle depolarization ratio of one aerosol ($\delta_x$), as well as the particle linear depolarization ratio of the mixture ($\delta_p$), at a given wavelength (Eq. 7). For two aerosol types and depolarization ratios at two wavelengths, four $Q_x(\lambda_i)$ are defined and considered known based on prior information or reasonable assumption. Equation 5 can be thus simplified as Eq. 8. Equations 6 and 8 form a system with two unknowns, yielding analytical expressions of $\phi_a$ and $\phi_b$ as shown in Eqs. 9 and 10. These expressions are equivalent to Eq. 14 presented in Tesche et al. (2009).

$$Q_x(\lambda_i) = \frac{\delta_p(\lambda_i) - \delta_x(\lambda_i)}{\delta_x(\lambda_i) + 1} \tag{7}$$

$$\phi_a(\lambda_i)Q_a(\lambda_i) + \phi_b(\lambda_i)Q_b(\lambda_i) = 0 \tag{8}$$

$$\phi_a(\lambda_i) = \frac{-Q_b(\lambda_i)}{Q_a(\lambda_i) - Q_b(\lambda_i)} \tag{9}$$

$$\phi_b(\lambda_i) = \frac{Q_a(\lambda_i)}{Q_a(\lambda_i) - Q_b(\lambda_i)} \tag{10}$$

Using Eqs. 3 (for wavelengths $\lambda_1$ and $\lambda_2$) and 4 (for aerosol types $a$ and $b$), the relationship among the aerosol backscatter fractions at two wavelengths can be derived as Eqs. 11 and 12. Together with the two expressions in Eq. 5, corresponding to wavelengths $\lambda_1$ and $\lambda_2$, these four equations enable the derivation of the mathematical relationship between $\delta_p(\lambda_1)$ and $\delta_p(\lambda_2)$, as expressed in Eq. 13, based on six variables ($\delta_a(\lambda_1)$, $\delta_b(\lambda_1)$, $\delta_a(\lambda_2)$, $\delta_b(\lambda_2)$, $\mathring{A}_{\beta,a}(\lambda_1,\lambda_2)$, $\mathring{A}_{\beta,b}(\lambda_1,\lambda_2)$). Prior knowledge of these values is essential, and examples will be discussed later. Such a relationship relating to the wavelength dependency on the particle linear depolarization ratios is fixed for the mixture of two aerosol types and can be represented as a curve. The two endpoints of the curve are determined by the $\delta_x$ of the two aerosols, while $\mathring{A}_{\beta,x}$ of both aerosol types impacts its curvature.

$$\phi_a(\lambda_1) = \frac{\eta_a(\lambda_1,\lambda_2)\phi_a(\lambda_2)}{\eta_a(\lambda_1,\lambda_2)\phi_a(\lambda_2) + \eta_b(\lambda_1,\lambda_2)\phi_b(\lambda_2)} \tag{11}$$

$$\phi_b(\lambda_1) = \frac{\eta_b(\lambda_1,\lambda_2)\phi_b(\lambda_2)}{\eta_a(\lambda_1,\lambda_2)\phi_a(\lambda_2) + \eta_b(\lambda_1,\lambda_2)\phi_b(\lambda_2)} \tag{12}$$

$$\delta_p(\lambda_1) = \frac{\delta_a(\lambda_1)\eta_a[\delta_b(\lambda_1)+1][\delta_a(\lambda_2)+1][\delta_b(\lambda_2)-\delta_p(\lambda_2)] + \delta_b(\lambda_1)\eta_b[\delta_a(\lambda_1)+1][\delta_b(\lambda_2)+1][\delta_p(\lambda_2)-\delta_a(\lambda_2)]}{\eta_a[\delta_b(\lambda_1)+1][\delta_a(\lambda_2)+1][\delta_b(\lambda_2)-\delta_p(\lambda_2)] + \eta_b[\delta_a(\lambda_1)+1][\delta_b(\lambda_2)+1][\delta_p(\lambda_2)-\delta_a(\lambda_2)]} \tag{13}$$

It is concluded that the relationship between $\delta_p$ at two wavelengths is not linear. Thus, the commonly used ratio of $\delta_p(\lambda_1)$ and $\delta_p(\lambda_2)$, which is typically assumed to be constant, is less accurate than the characteristic curved relationship proposed in this study. Furthermore, these characteristic wavelength dependencies provide the potential application for aerosol typing.

## 2.2 Mixture of three aerosol types

As a following step, we consider more complicated aerosol mixtures, consisting of three aerosol types (denoted with the subscript $a$, $b$ or $c$). The particle depolarization ratios ($\delta_x$) of all three types should be different and known.

Using Eqs. 2 and 3, $\delta_p$ of the particle ensemble at wavelength $\lambda_1$ (or $\lambda_2$) can be expressed by Eq. 14 (or 15). The sum of $\phi_a(\lambda_i)$, $\phi_b(\lambda_i)$, and $\phi_c(\lambda_i)$ is clearly equal to 1 for $i=1$ and $i=2$ (Eqs. 16–17). The relationships among the aerosol backscatter fractions at two wavelengths can be derived as Eqs. 18–20, and remain valid under the interchange of $\lambda_1$ and $\lambda_2$. Note that any one of Eqs. 17–20 can be derived from the other three. Therefore, among these four equations, there are only three independent equations.

$$\delta_p(\lambda_1) = \frac{\frac{\phi_a(\lambda_1)\delta_a(\lambda_1)}{\delta_a(\lambda_1)+1} + \frac{\phi_b(\lambda_1)\delta_b(\lambda_1)}{\delta_b(\lambda_1)+1} + \frac{\phi_c(\lambda_1)\delta_c(\lambda_1)}{\delta_c(\lambda_1)+1}}{\frac{\phi_a(\lambda_1)}{\delta_a(\lambda_1)+1} + \frac{\phi_b(\lambda_1)}{\delta_b(\lambda_1)+1} + \frac{\phi_c(\lambda_1)}{\delta_c(\lambda_1)+1}} \tag{14}$$

$$\delta_p(\lambda_2) = \frac{\frac{\phi_a(\lambda_2)\delta_a(\lambda_2)}{\delta_a(\lambda_2)+1} + \frac{\phi_b(\lambda_2)\delta_b(\lambda_2)}{\delta_b(\lambda_2)+1} + \frac{\phi_c(\lambda_2)\delta_c(\lambda_2)}{\delta_c(\lambda_2)+1}}{\frac{\phi_a(\lambda_2)}{\delta_a(\lambda_2)+1} + \frac{\phi_b(\lambda_2)}{\delta_b(\lambda_2)+1} + \frac{\phi_c(\lambda_2)}{\delta_c(\lambda_2)+1}} \tag{15}$$

$$\phi_a(\lambda_2) + \phi_b(\lambda_2) + \phi_c(\lambda_2) = 1 \tag{16}$$

$$\phi_a(\lambda_1) + \phi_b(\lambda_1) + \phi_c(\lambda_1) = 1 \tag{17}$$

$$\phi_a(\lambda_1) = \frac{\eta_a(\lambda_1,\lambda_2)\phi_a(\lambda_2)}{\eta_a(\lambda_1,\lambda_2)\phi_a(\lambda_2) + \eta_b(\lambda_1,\lambda_2)\phi_b(\lambda_2) + \eta_c(\lambda_1,\lambda_2)\phi_c(\lambda_2)} \tag{18}$$

$$\phi_b(\lambda_1) = \frac{\eta_b(\lambda_1,\lambda_2)\phi_b(\lambda_2)}{\eta_a(\lambda_1,\lambda_2)\phi_a(\lambda_2) + \eta_b(\lambda_1,\lambda_2)\phi_b(\lambda_2) + \eta_c(\lambda_1,\lambda_2)\phi_c(\lambda_2)} \tag{19}$$

$$\phi_c(\lambda_1) = \frac{\eta_c(\lambda_1,\lambda_2)\phi_c(\lambda_2)}{\eta_a(\lambda_1,\lambda_2)\phi_a(\lambda_2) + \eta_b(\lambda_1,\lambda_2)\phi_b(\lambda_2) + \eta_c(\lambda_1,\lambda_2)\phi_c(\lambda_2)} \tag{20}$$

Thus, it is possible to retrieve the six unknowns $\phi_x(\lambda_i)$ (with $x$ being $a$, $b$, or $c$, and $i$ being 1 or 2) using the six independent equations (Eqs. 14–20) with a unique mathematical solution. Equation 14 or 15 can be simplified as Eq. 21 using the aerosol-type-dependent term $Q_x(\lambda_i)$ (Eq. 7). The expressions of the aerosol backscatter fraction $\phi_x(\lambda_i)$ can be mathematically derived using Eqs. 16–21. An example of the expression of $\phi_a(\lambda_2)$ solution is given in Eq. 22. The solution equations for the other two types can apply the same equation, with the indices $a$, $b$, and $c$ interchanged accordingly. The aerosol backscatter coefficient of one type can be easily calculated from the total backscatter coefficient using the backscatter fraction.

$$\phi_a(\lambda_i)Q_a(\lambda_i) + \phi_b(\lambda_i)Q_b(\lambda_i) + \phi_c(\lambda_i)Q_c(\lambda_i) = 0 \tag{21}$$

$$\phi_a(\lambda_2) = \frac{\eta_b Q_b(\lambda_1)Q_c(\lambda_2) - \eta_c Q_c(\lambda_1)Q_b(\lambda_2)}{\eta_a Q_a(\lambda_1)[Q_b(\lambda_2) - Q_c(\lambda_2)] + \eta_b Q_b(\lambda_1)[Q_c(\lambda_2) - Q_a(\lambda_2)] + \eta_c Q_c(\lambda_1)[Q_a(\lambda_2) - Q_b(\lambda_2)]} \tag{22}$$

## 2.3 Synthetic aerosol mixture

In this section, we apply the algorithm for the decomposition of two or three aerosol types to synthetic mixtures for validating the methodology. Dust mixture is a good candidate, since the optical properties of fine and coarse dust have been measured and reported in many laboratory and field studies (Sakai et al., 2010; Järvinen et al., 2016; Freudenthaler et al., 2009; Miffre et al., 2016; Burton et al., 2015), and they present distinct particle depolarization ratios. Furthermore, atmospheric mineral dust is one of the most important aerosol types, playing a key role on several critical aspects of the Earth system. The size of mineral dust particles in the atmosphere ranges from less than 0.1 $\mu$m to more than 100 $\mu$m in diameter (Adebiyi et al., 2023; Ryder et al., 2019; van der Does et al., 2018; Mahowald et al., 2014). Similar to POLIPHON method, we consider dust particles in the size spectrum up to 1 $\mu$m in diameter as fine mode, and coarse-mode dust in the super-micrometer size spectrum (>1 $\mu$m in diameter).

From the method descriptions (Sects. 2.1–2.2), it becomes evident that adequate knowledge of the depolarization ratios ($\delta_x$) and backscatter-related Ångström exponent ($\AA_{\beta,x}$) of each aerosol type is necessary. These input parameters are summarized and/or assumed in Table 1. The particle depolarization ratios of coarse and fine dust at 355, 532, and 1064 nm are taken from the review study of Mamouri and Ansmann (2017) (see Table 1 in that paper), which includes two laboratory studies (Sakai et al., 2010; Järvinen et al., 2016), and five field observations (Freudenthaler et al., 2009; Burton et al., 2015; Veselovskii et al., 2016; Haarig et al., 2017a; Hofer et al., 2017).

The laboratory experiments to measure the airborne dust properties are challenging. Sakai et al. (2010) measured $\delta_x(532)$ of several tropospheric aerosols using a laboratory chamber. For high number concentrations, $\delta_{df}(532)$ of the fine dust were found to be 0.17 ± 0.03 for the Asian dust and 0.14 ± 0.03 for the Saharan dust, whereas $\delta_{dc}(532)$ of the coarse dust were 0.39 ± 0.04 to 0.05 for both. Järvinen et al. (2016) measured $\delta_{dust}$ of various dust samples at 488 and 552 nm in a cloud chamber. They reported that the measured $\delta_{dust}$ ranged from 0.03 to 0.36 and were strongly dependent on the particle size. Based on these studies, Mamouri and Ansmann (2017) estimated $\delta_{df}$ for fine dust of around 0.21 ± 0.02, 0.16 ± 0.02, and 0.09 ± 0.03 for the laser wavelengths of 355, 532, and 1064 nm, respectively. Miffre et al. (2023) found that the dust depolarization ratios are mainly influenced by the particles' complex refractive index, when the strongly light-absorbing hematite is present, while its variations with size and shape are less significant. They also present measured $\delta_{dust}$ values for finer and coarser size distributions of Arizona and Asian dust (see Table 1 in Miffre et al., 2023), which can not be used in this study as the used size distributions are different from other studies.

Field measurements were also considered. Nevertheless, the reported particle depolarization ratios from field observations may still contain contributions from spherical particles within the mixture, potentially leading to an underestimation of the characteristic values. Moreover, certain vertical smoothing is always applied in the lidar retrievals, and layer-mean values are often reported, adding uncertainties on the lidar-derived characteristics of aerosol types. Burton et al. (2015) performed airborne measurements of dust plumes over the United States. For a dense dust layer of local North American dust, they found high $\delta_p$ of 0.24 ± 0.05, 0.37 ± 0.01, 0.38 ± 0.01 at 355, 532, and 1064 nm, respectively. The corresponding $\AA_{\beta,p}(532,1064)$ was -0.09 ± 0.04. For two transported Saharan dust layers, $\AA_{\beta,p}(532,1064)$ were found to be 0.46 ± 0.03 and 0.68 ± 0.13. Veselovskii et al. (2016) observed that $\delta_p(532)$ increased up to 0.35 ± 0.05, and $\AA_{\beta,p}(355,532)$ decreased to -0.7, during strong African dust episodes. Hofer et al. (2017) present an extreme dust event (probably with coarse dust dominant) with values of 0.29 ± 0.01 or 0.35 ± 0.01 for $\delta_p(355)$ or $\delta_p(532)$, and values of -0.20 ± 0.13 or 0.29 ± 0.03 for $\AA_{\beta,p}(355,532)$ or $\AA_{\beta,p}(532,1064)$, respectively. Freudenthaler et al. (2009) reported 0.31 ± 0.03 and 0.27 ± 0.04 for $\delta_p(532)$ and $\delta_p(1064)$ over Morocco. Haarig et al. (2017a) highlighted maximum values of 0.27 for $\delta_p(1064)$ measured over Barbados, which are almost equal to the coarse dust depolarization ratio.

The characteristics of non-dust type can vary widely across regions and times. A review study of Proestakis et al. (2024) suggests a value of 0.05 ± 0.02 at 532 nm for non-dust depolarization ratios (Tesche et al., 2009; Mamouri and Ansmann, 2014, 2016; Marinou et al., 2017; Proestakis et al., 2018). Here we consider 0.05 ± 0.02 for all 3 wavelengths in the synthetic examples and cases where measurements are unavailable. However, such values may be adjusted if measurements become available.

**Table 1.** Input parameters used in the synthetic aerosol mixtures (Sects. 2.3–2.4) and case studies (Sect. 3): particle depolarization ratios ($\delta_x$) at 355, 532 or 1064 nm, and backscatter-related Ångström exponent ($\mathring{A}_{\beta,x}$) between 355 and 532 nm, or between 532 and 1064 nm, for coarse dust (dc), fine dust (df), and spherical non-dust (nd) aerosols. Typical values are considered from published studies (see text in Sect. 2.3). Values in parentheses are based on our assumptions.

| Aerosol type (abbreviation) | $\delta_x(355)$ | $\delta_x(532)$ | $\delta_x(1064)$ | $\mathring{A}_{\beta,x}(355, 532)$ | $\mathring{A}_{\beta,x}(532, 1064)$ |
|---|---|---|---|---|---|
| Coarse dust (dc) | $0.27 \pm 0.03$ | $0.37 \pm 0.03$ | $0.27 \pm 0.03$ | $-0.20\ (\pm 0.03)$ | $0.30 \pm 0.03$ |
| Fine dust (df) | $0.21 \pm 0.02$ | $0.16 \pm 0.02$ | $0.09 \pm 0.02$ | $1.50\ (\pm 0.03)$ | $0.60\ (\pm 0.03)$ |
| Non-dust aerosol (nd) | $0.05 \pm 0.02$ | $0.05 \pm 0.02$ | $0.05 \pm 0.02$ | $2.00\ (\pm 0.03)$ | $(1.50 \pm 0.03)$ |

Laboratory measurements of dust $\mathring{A}_\beta$ are still missing, thus values from lidar measurements are considered. Mamouri and Ansmann (2014) stated the extinction-related Ångström exponent between 355 and 532 nm to be -0.2, 1.5, 2.0 for coarse dust, fine dust, and non-dust aerosols, respectively. Assuming the same lidar ratios at 355 and 532 nm, these values can be used for the $\mathring{A}_{\beta,x}(355, 532)$. The lidar ratio, defined as the extinction-to-backscatter ratio, has been widely used in lidar-based aerosol classification algorithms because it provides information on aerosol type. Numerous lidar studies have investigated the spectral dependence of the lidar ratio for different aerosol types (e.g., Haarig et al., 2025). For instance, Floutsi et al. (2023) present a comprehensive collection of depolarization ratios, lidar ratios, and Ångström exponents for different aerosol types and mixtures based on ground-based lidar observations. For most aerosol types, including dust from most regions except Central Asia, the assumption of lidar ratio equality between 355 and 532 nm is generally valid within observational uncertainties. However, for smoke mixtures, this assumption should be applied with caution. The value for $\mathring{A}_{\beta,df}(532, 1064)$ for the fine dust was not found from the literature. Nevertheless, several studies documented range of values of $\mathring{A}_{\beta,p}$ for the dust layers including the coarse and fine dust. For example, Tesche et al. (2009) and Filioglou et al. (2020) found $\mathring{A}_{\beta,p}(532, 1064)$ values of 0.0 to 0.7, or 0.1 to 0.6, respectively. The upper limit of such range values could be related to the fine dust dominant dust mixture.

It should be emphasized that this paper focuses on presenting an easy-to-apply methodology, and the investigation of the optical properties of individual aerosol types is beyond the scope. The optical properties of the various aerosol types can be readily updated as more accurate values become available and new observations emerge in the field.

To apply the proposed method, we begin by simulating a synthetic mixture of coarse dust and spherical non-dust aerosol, using the input parameters from Table 1. The particle depolarization ratios of the mixture can be easily derived for different coarse dust backscatter fractions (Eq. 5 and Fig. 1a). Then, a scatter plot using $\delta_p$ at two wavelengths can be generated, and linked to the backscatter fractions (Fig. 1b). As mentioned in Sect. 2.1, such a relationship depends on the optical properties of each aerosol type. For the synthetic example in Fig. 1(b-d), the top-right (or left-bottom) endpoint of the curve is determined by the pair values of $\delta_{dc}$ (or $\delta_{nd}$) at 355 and 532 nm. Also, examples of characteristic curves using different $\mathring{A}_{\beta,x}(355, 532)$ values are given in Fig. 1(d) to illustrate the curvature effect: higher $\mathring{A}_{\beta,nd}$ or lower $\mathring{A}_{\beta,dc}$ will result in a higher curvature while bending towards the right bottom direction, and vice versa. A sensitively study on the synthetic example (Fig. 1) was performed based on the Monte Carlo approach. Six variables ($\delta_{dc}(355), \delta_{dc}(532), \mathring{A}_{\beta,dc}(355, 532), \delta_{nd}(355), \delta_{nd}(532), \mathring{A}_{\beta,nd}(355, 532)$)

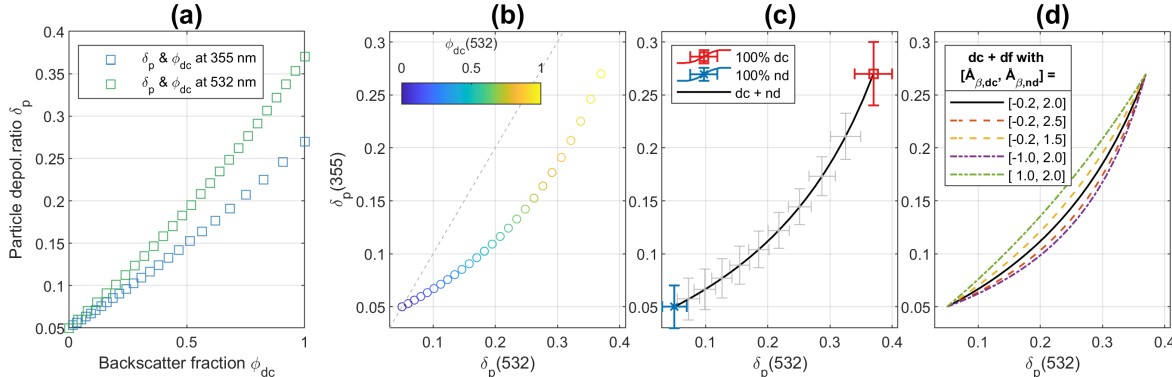

**Figure 1.** Synthetic particle linear depolarization ratios ($\delta_p$) for the aerosol mixture with two aerosol types: coarse dust (dc) and spherical non-dust aerosol (nd). (a) $\delta_p$ against the coarse dust backscatter fraction ($\phi_{dc}$) at 355 or 532 nm. (b) Relationship between $\delta_p$ at 355 and 532 nm, with $\phi_{dc}$ at 532 nm shown by color scales. (c) Theoretical characteristic curves of $\delta_p$ at 355 and 532 nm for the dc and nd mixture, with uncertainties from Monte Carlo simulations (using parameters in Table 1). (d) Characteristic curves of $\delta_p$ at 355 and 532 nm for dc and nd mixture, with different input values for backscatter-related Ångström exponents ($\text{Å}_{\beta,dc}(355,532)$ and $\text{Å}_{\beta,nd}(355,532)$).

are used, considering normal statistical distribution with their standard derivations provided in Table 1. Results using 1000 draws/simulations are shown as gray error bars in Fig. 1(c).

Next, we consider a synthetic mixture of three aerosol types (coarse dust, fine dust, and spherical non-dust aerosol), using the input parameters provided in Table 1. $\delta_p$ value of the mixture depends on the share of each aerosol type (Eqs. 14–15 and Fig. 2a). For any combination of fractions (each ranging from 0 to 1), the pair values of $\delta_p$ at two wavelengths must remain within the enclosed region predetermined by three boundary curves (Fig. 2b), and each curve is determined by the optical properties of any two of three types (e.g., using Eq. 13, see Fig. 1b-d). Similar sensitively study was performed based on the Monte Carlo approach, with results shown in Fig. 2(c), using the optical properties and uncertainties of $\delta_x$ and $\text{Å}_{\beta,x}$ in Table 1.

## 2.4 Uncertainty study

Decomposition of two aerosol types is well studied and reported in many studies. In this section, we present a sensitivity study and explore the uncertainties of the proposed methodology for the decomposition of three aerosol types using particle depolarization ratios measured at two wavelengths.

Synthetic data representing aerosol mixtures, with different fractions of coarse dust, fine dust, and spherical non-dust aerosol, are utilized with input parameters from Table 1. Two wavelengths 355 and 532 nm are considered. Hereafter, if $\text{Å}_{\beta,x}$ is mentioned without specifying wavelengths, it refers to $\text{Å}_{\beta,x}(355,532)$. As input variables, there are two observational parameters ($\delta_p(355)$ and $\delta_p(532)$) for the mixture, and nine type-specific optical properties ($\delta_{dc}(355)$, $\delta_{dc}(532)$, $\text{Å}_{\beta,dc}$, $\delta_{df}(355)$, $\delta_{df}(532)$, $\text{Å}_{\beta,df}$, $\delta_{nd}(355)$, $\delta_{nd}(532)$, $\text{Å}_{\beta,nd}$). Under ideal noise-free conditions, the measured pair values of $\delta_p$ of the aerosol mixture, with whichever the mixing ratio of the three types, should be inside the region bounded by the three characteristic curves.

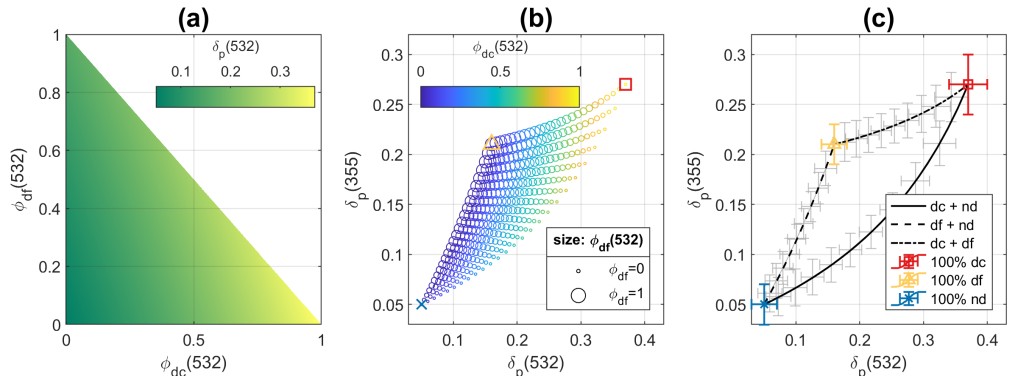

**Figure 2.** Synthetic particle linear depolarization ratios ($\delta_p$) of the particle ensemble at 355 and 532 nm for the aerosol mixture with three aerosol types: coarse dust (dc), fine dust (df), and spherical non-dust aerosol (nd). (a) $\delta_p$ at 532 nm for different backscatter fractions ($\phi$) of coarse and fine dust. (b) $\delta_p$ at 355 and 532 nm, with coarse dust backscatter fractions ($\phi_{dc}$) shown by color scales, and fine dust backscatter fractions ($\phi_{df}$) shown by marker sizes. (c) Theoretical characteristic curves of $\delta_p$ at 355 and 532 nm for the two aerosol mixtures are in black lines, with uncertainties from Monte Carlo simulations (using parameters in Table 1). The pair values of $\delta_{dc}$, $\delta_{df}$, and $\delta_{nd}$ is shown as red square, yellow triangle, or blue cross, respectively.

Three cases are manually selected to illustrate different mixing conditions, with initial $\delta_p$ values given in Table 2 and Fig. 3(a). Note that it is purely a mathematical problem, as the system can be solved regardless of whether the values are physically meaningful. For example, the pair values of $\delta_p$ of case 3 locate outside the characteristic region, resulting in negative value of $\phi_{df}(532)$ and above one value of $\phi_{dc}(532)$. The results of the sensitivity study depend on the initial values of the measurable $\delta_p(355)$ and $\delta_p(532)$, and vary from case to case. In this section, we present the sensitivity study based on these three selected cases. We first perform a global sensitivity analysis to assess the combined effect of all input variables, followed by an individual sensitivity analysis to evaluate the impact of each variable separately. In addition, we investigate the influence of the uncertainty levels of each input variable. Furthermore, an additional analysis is conducted to determine the tolerated bias in observational parameters. These analyses are carried out using Monte Carlo simulations.

In the first step, we consider a 5 % uncertainty on $\delta_p$ at both 355 and 532 nm. This is in the lower end of typical observational relative uncertainty of 5–10 % (e.g., Baars et al., 2012; Tesche et al., 2009; Freudenthaler et al., 2009). The type-specific optical properties are assumed with uncertainties given in Table 1. The aerosol backscatter fractions can be calculated using Eq. 22. The ideal results without introducing uncertainties are given as "Reference results" in Table 2. A Monte Carlo simulation was conducted involving 10 000 draws based on normal distributions of 11 variables. For each draw, the aerosol backscatter fractions of three types are calculated, resulting the distributions shown in Fig. 3(b-d). The statistical parameters are given in Table 2, where skewness describes the asymmetry of a distribution, and kurtosis describes how peaked or flat it is relative to a normal distribution. The retrieved $\phi_x$ distributions are also presented as ternary plots in Fig. 3(e), where the color scale represent the frequency of occurrences. We can easily identify areas where there is a greater reliability for the $\phi_x$ retrievals with

**Table 2.** Parameters for the uncertainty study. Three cases with different initial particle linear depolarization ratios ($\delta_p$) of the particle ensemble at 355 and 532 nm are used. Reference results are calculated using Eq. 22 without introducing uncertainties. Statistical parameters of retrieved aerosol backscatter fraction $\phi_x$ of coarse dust (dc), fine dust (df), and non-dust aerosol (nd) are given after Monte Carlo simulations.

| | Initial $\delta_p$ | | Reference results | | Monte Carlo results | | | |
| --- | --- | --- | --- | --- | --- | --- | --- | --- |
| | $\delta_p(355)$ | $\delta_p(532)$ | | | Mean | Standard deviation | Skewness | Kurtosis |
| Case 1 | 0.16 | 0.19 | $\phi_{dc}(532)$ | 0.33 | 0.33 | 0.11 | 0.22 | 8.48 |
| | | | $\phi_{df}(532)$ | 0.42 | 0.42 | 0.17 | -1.04 | 32.18 |
| | | | $\phi_{nd}(532)$ | 0.25 | 0.25 | 0.08 | 2.38 | 74.78 |
| Case 2 | 0.18 | 0.28 | $\phi_{dc}(532)$ | 0.74 | 0.76 | 0.16 | 1.01 | 33.62 |
| | | | $\phi_{df}(532)$ | 0.08 | 0.05 | 0.22 | -0.54 | 54.88 |
| | | | $\phi_{nd}(532)$ | 0.19 | 0.19 | 0.08 | -0.33 | 60.15 |
| Case 3 | 0.10 | 0.30 | $\phi_{dc}(532)$ | 1.01 | 1.05 | 0.42 | 59.49 | 4817.73 |
| | | | $\phi_{df}(532)$ | -0.46 | -0.52 | 0.64 | -56.71 | 4297.81 |
| | | | $\phi_{nd}(532)$ | 0.45 | 0.47 | 0.26 | 39.65 | 2295.32 |

darker color. For case 1, the pair values of $\delta_p$ locate in the middle of the enclosed characteristic region. The retrieved backscatter fractions are closer in shape to a Gaussian distribution than the other two cases. Their values are reasonably between 0 to 1. $\delta_p(355)$ and $\delta_p(532)$ of case 2 are located inside the error bars of the characteristic curve of coarse dust and non-dust aerosol. Thus, for this $\delta_p$ pair, it is possible that the mixture contains only coarse dust and non-dust aerosol, or contains three aerosol types. For example, a small part of simulated values of $\phi_{df}$ distribution in Fig. 3(c) is negative due to the uncertainties. In case 3, the pair values of $\delta_p$ are far away from the characteristic region, only un-physical values (always below 0) were derived for $\phi_{df}$ (Fig. 3d), and a big portion of $\phi_{dc}$ are bigger than 1. This case is clearly not a mixture of above mentioned three aerosol types. It could contain other depolarizing particles, e.g., pollen, in the mixture.

As a second step, we perform an individual sensitivity analysis using the One-at-a-Time (OAT) method, to assess the influence of each input variable independently. For each variable, 10 000 Monte Carlo simulations are conducted based on normal statistical distributions, using the mean and standard deviation specified in the previous step, while all other variables remain fixed. From these simulations, we derive $\phi_x(532)$ with its mean and standard deviation. The resulting uncertainties are presented in Fig. 4, illustrating the impact of each variable under uncertainty. $\mathring{A}_{\beta,dc}$, $\mathring{A}_{\beta,df}$, and $\mathring{A}_{\beta,nd}$ were found to have only minor effects on $\phi_x(532)$ across all cases, with a largest uncertainty of 5 % on $\phi_{df}(532)$ attributed to $\mathring{A}_{\beta,df}$ in case 1. For all three cases, $\delta_{dc}(532)$ and $\delta_p(532)$ show strong influence on $\phi_{dc}(532)$ and $\phi_{df}(532)$. In addition, $\delta_{dc}(355)$, $\delta_{nd}(355)$, and $\delta_p(355)$ also show notable influence on $\phi_{df}(532)$. In case 2, the optical properties of fine dust ($\delta_{df}(355)$, $\delta_{df}(532)$, and $\mathring{A}_{\beta,df}$) have small impacts on $\phi_x(532)$. Because the pair values of $\delta_p$ are located near the characteristic curve of coarse dust and non-dust aerosols, the contribution from fine dust is consequently reduced. The relative uncertainties on $\phi_{df}(532)$ in case 2 are large due to its small value (Fig. 4e).

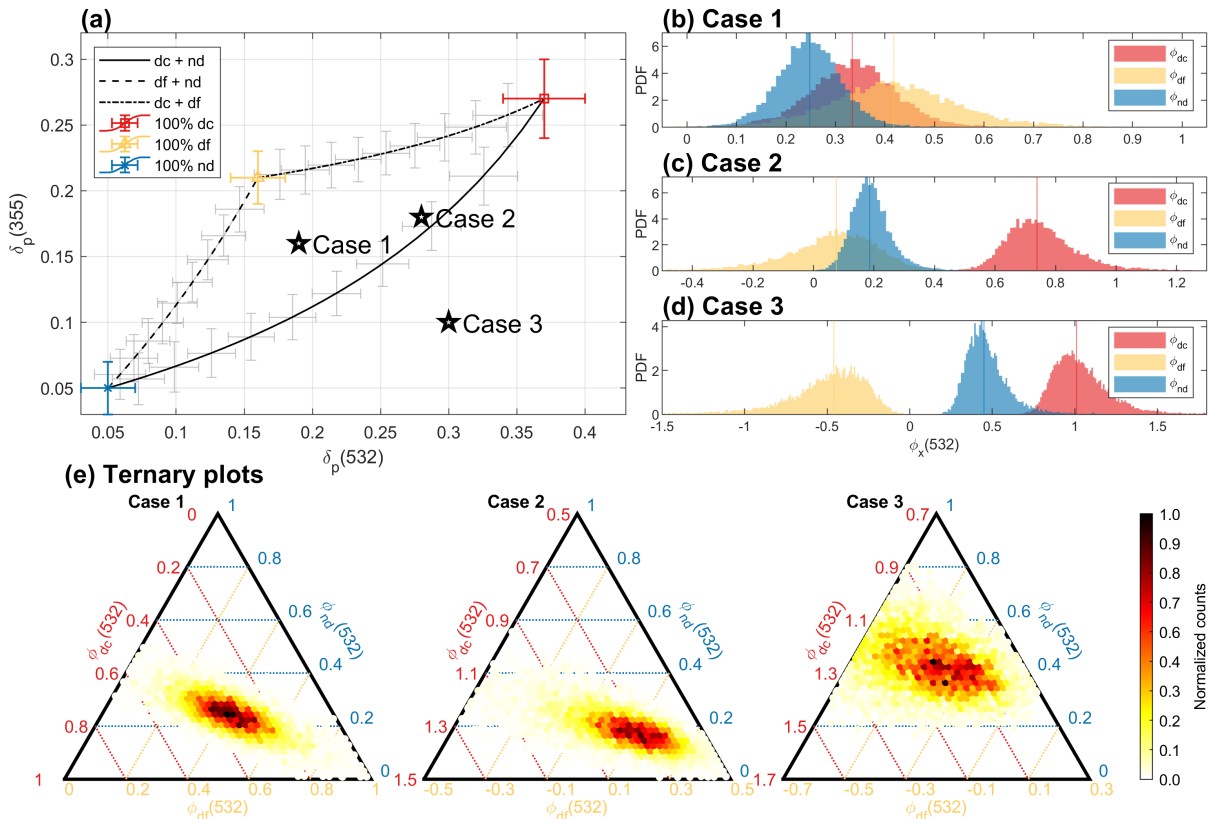

**Figure 3.** (a) Theoretical characteristic curves of $\delta_p$ at 355 and 532 nm for the 2 aerosol mixtures are in black, with uncertainties from Monte Carlo simulations (using parameters in Table 1). The pair values of $\delta_x$ for coarse dust (dc), fine dust (df), or spherical non-dust (nd) aerosol are shown as red square, yellow triangle, or blue cross, respectively. Initial $\delta_p$ pairs of three cases are shown by black stars. (b-d) Probability density function (PDF) estimates of the retrieved backscatter fractions ($\phi_x$) of dc, df, and nd aerosol, for each case. Monte Carlo approach was applied. (e) Ternary plots of retrieved $\phi_x$ for 3 cases, with normalized counts shown as color.

To further investigate how the uncertainty level of each individual input variable influence the results, we extend the OAT method by varying the relative uncertainty of each input variable from 1 % to 99 %. For each uncertainty level, 10 000 Monte Carlo simulations are conducted. We continue to analyze the three previously defined cases. For case 2, we exclude the resulting uncertainties on $\phi_{df}$ from the discussion due to its large relative uncertainty as mentioned earlier. For each individual input variable, the resulting uncertainty in $\phi_x$ increases as the uncertainty in that input variable increases; however, the rate

of increase differs among variables. The resulting maximum allowable relative uncertainty per variable to ensure relative uncertainty of retrieved backscatter fractions below 10 %, 20 %, or 50 % is given in Fig. 5. A larger maximum allowable relative uncertainty on the variable (longer bar) indicates lower sensitivity. We first consider the nine type-specific optical properties. Our analysis indicates that $Å_{\beta,dc}$, $Å_{\beta,df}$, $Å_{\beta,nd}$, and $\delta_{nd}(532)$ have minor influences on $\phi_x(532)$. Their uncertainties can reach up to almost 100 % without exceeding the 50 % on $\phi_x(532)$'s uncertainty. Among the remaining variables, $\delta_{nd}(355)$ exhibits

the lowest sensitivity. In contrast, $\delta_{dc}(532)$ and $\delta_{df}(355)$ are the most critical ones, as their relative uncertainties must remain below $\sim$15 % to the keep the relative uncertainties on $\phi_x(532)$ below 50 %. Followed by $\delta_{dc}(355)$ and $\delta_{df}(532)$, which also show substantial impacts. Next, we found that the observational uncertainties on $\delta_p(355)$ and $\delta_p(532)$ significantly affects the uncertainty on the retrieved $\phi_x(532)$. In case 1, the relative uncertainties on $\delta_p(355)$ and $\delta_p(532)$ should be under 17 % to keep $\phi_x(532)$'s uncertainties below 50 %. To achieve a more accurate retrieval, like, for example by reducing the uncertainties associated with $\phi_x(532)$ to below 20 % (or 10 %), the relative uncertainties of $\delta_p(355)$ and $\delta_p(532)$ must be limited to less than 7 % (or 4 %).

An additional analysis was carried out to investigate the tolerated bias on observational parameters $\delta_p$. Different $\delta_p(355)$ or $\delta_p(532)$ values were applied independently to compute the corresponding $\phi_x(532)$. An example of results for case 1 is given in Fig. 6. A maximum bias of 0.01 (or 0.03) can be tolerated for the measured $\delta_p(355)$ and $\delta_p(532)$ to keep the uncertainties on $\phi_x(532)$ to below 20 % (or 50 %). Hence, there is the demand of well characterized lidar systems to deliver high-quality depolarization ratios.

# 3  Case studies of mineral dust

In this section, the algorithm was applied to lidar observations of aerosols in different dust-affected regions.

## 3.1  Arabian dust

Filioglou et al. (2020) reported the aerosol particle properties of Arabian dust over a rural site in the United Arab Emirates, during the 1 year measurement campaign between March 2018 and February 2019, within the framework of the Optimization of Aerosol Seeding In rain enhancement Strategies (OASIS) project. Among 1130 night-time aerosol particle layers, $\delta_p$ at both 355 and 532 nm are available for 1063 layers. These layer-mean $\delta_p$ are plotted in Fig. 7, with center mass altitude of the layers shown by color scale, and layer depths shown by marker sizes.

The measurement site is a receptor of frequent dust events, three possible aerosol types are assumed to be present in these layers: coarse and fine dust (with optical properties in Table 1), and the spherical non-dust aerosols with anthropogenic and/or marine origin. From the lidar measurements, the mean values of the particle depolarization ratios for the non-dust aerosols ($\delta_{nd}$) are derived as $0.02 \pm 0.01$ at both 355 and 532 nm. $Å_{\beta,nd}(355,532)$ is assumed as 2 (cf., Table 1). The majority of the cases are well within the boundaries defined by the characteristic values of the three-aerosol-type method (Fig. 7). At higher altitudes, the $\delta_p$ pairs tend to lie closer to the characteristic curve of dc & nd, whereas layers at lower altitudes are located nearer to the characteristic curve of df & nd. This pattern suggests that dust with a higher altitude is generally coarser (mixed with non-dust particles), while dust with a lower altitude is finer. A few $\delta_p$ pairs are above the characteristic curve of dc & df. This could be an indication that the $\delta_x$ at 355 nm for pure Arabian dust (for both coarse and fine mode) may have slightly higher values. It may also be that the commonly reported pure values in the literature, based on field measurements, are often layer-mean values derived from smoothed optical parameters, which may underestimate the particle depolarization ratios of dust particles. These values were used as input for Table 1 in our calculations. To this direction, measuring the accurate optical properties

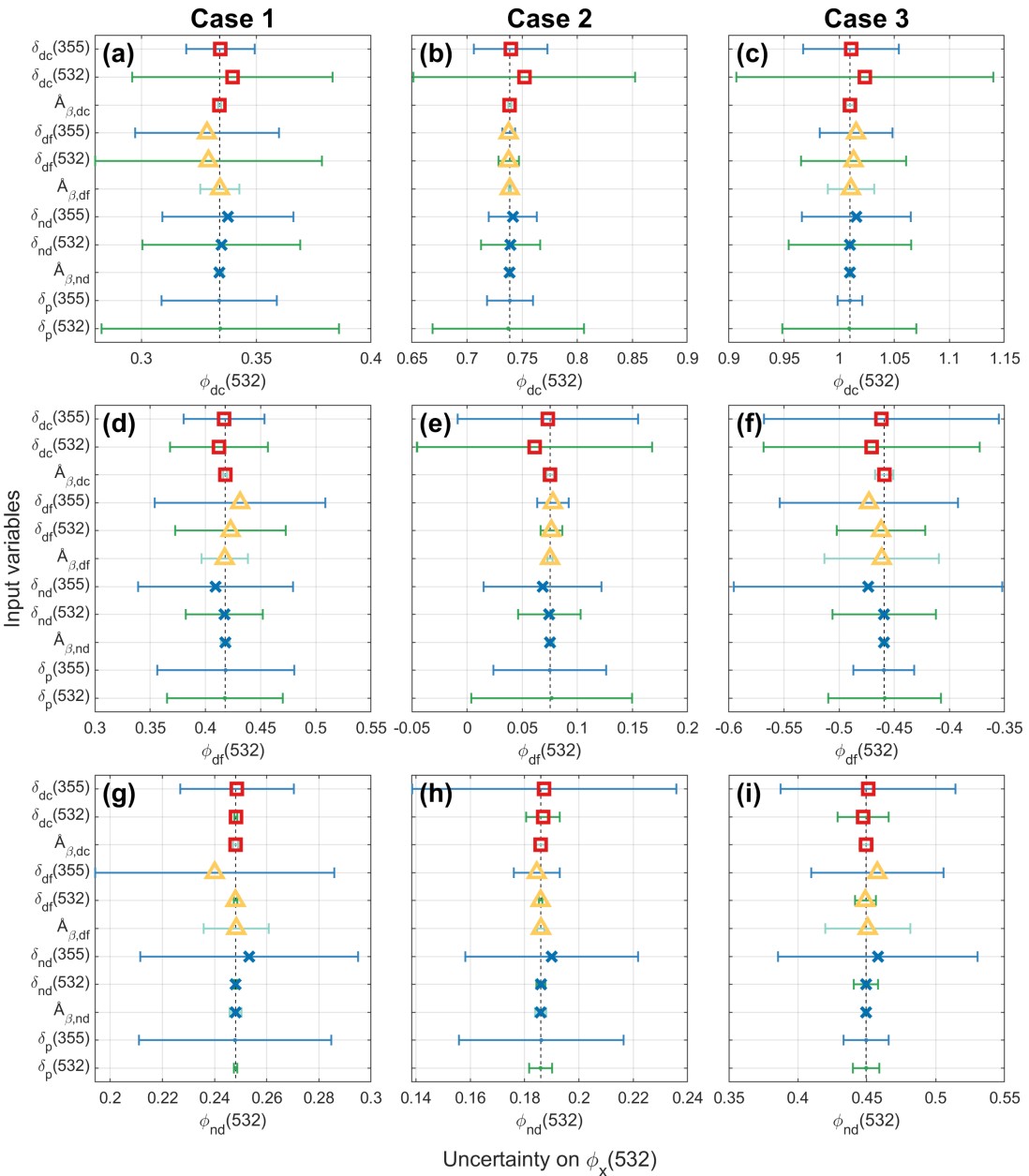

**Figure 4.** Estimated uncertainties on the retrieved backscatter fractions ($\phi_x$) at 532 nm of (a–c) coarse dust (dc), (d–f) fine dust (df), or (g–i) spherical non-dust (nd) aerosols, introduced by the uncertainty of each input variable, using the One-at-a-Time method. Colored markers represent the bias, whereas error bars indicate the standard deviation. Marker and error bar colors correspond to aerosol types and wavelengths, respectively, of individual input variables. Subplots of each column correspond to one case.

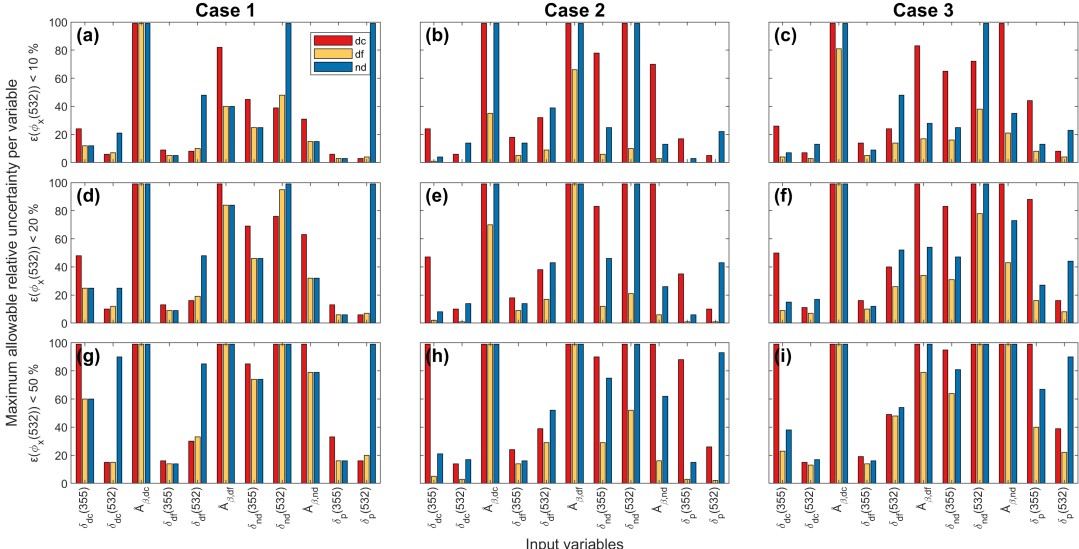

**Figure 5.** Maximum allowable relative uncertainty (values >100 % not shown) per variable to ensure relative uncertainty of retrieved backscatter fractions $\epsilon(\phi_x(532))$ are below 10 % (a-c), 20 % (d-f), or 50 % (g-i). An extended One-at-a-Time method has been utilized. Colors represent coarse dust (dc), fine dust (df), or spherical non-dust (nd) aerosols. Subplots of each column correspond to one case.

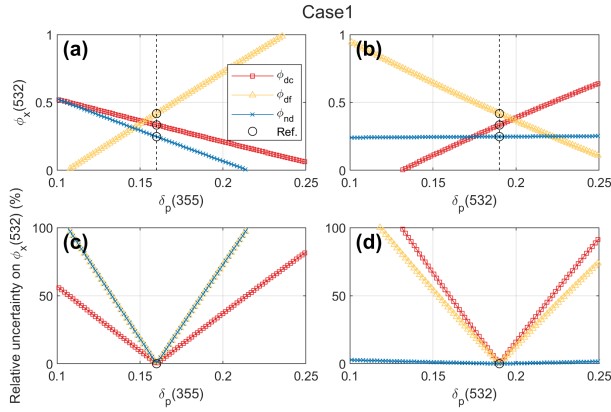

**Figure 6.** Estimated backscatter fractions ($\phi_x$) at 532 nm (a-b), and their relative uncertainties (c-d) of coarse dust (dc), fine dust (df), or spherical non-dust (nd) aerosols, against the applied different particle linear depolarization ratios at 355 or 532 nm ($\delta_p(355)$ or $\delta_p(532)$), for case 1. The reference values are shown as black circles.

of pure dust particles, e.g., in laboratory experiments or on the field, would be beneficial. Alternatively, the measurement data can serve as a prior information for determining particle depolarization ratios of each aerosol type. For instance, to include as many measured pairs as possible inside the characteristic region, we can assume $\delta_{dc}(355)$ and $\delta_{df}(355)$ as 0.31 and 0.23, respectively. At the same time, the curvature of the curves needs to be increased by raising $\mathring{A}_{\beta,nd}(355,532)$ to 2.5. The resulting

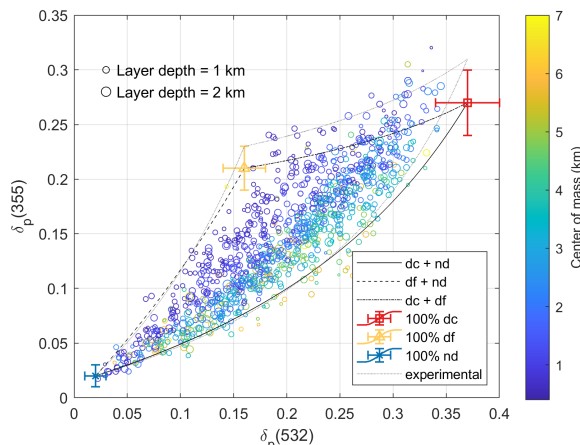

**Figure 7.** Scatter plot of layer-mean particle linear depolarization ratios ($\delta_p$) of the aerosol mixture at 355 and 532 nm of Arabian dust layers observed over United Arab Emirates. The altitude of the center of mass of the layers are shown by color scale, and layer depths shown by marker sizes. Theoretical characteristic curves of $\delta_p$ at 355 and 532 nm for the two aerosol mixtures are in black lines. Pair values of $\delta_x$ for coarse dust (dc), fine dust (df), or spherical non-dust (nd) aerosol are shown as red square, yellow triangle, or blue cross, respectively. Gray dotted lines show the characteristic curves derived from a preliminary adjustment based on experimental data.

characteristic region is indicated by gray dotted lines in Fig. 7. This adjustment appears to yield an improvement; however, it represents only a preliminary attempt and was not adopted in the subsequent analyses.

The aerosol backscatter fractions of the three aerosol types are calculated using Eq. 22. $\delta_p$ pairs located outside the region may result in negative or values above one for $\phi_x$. To avoid introducing a bias, these negative or above one values are included
in the analysis, as a large dataset is utilized. Final values of $\phi_x(532)$ are grouped using the center mass altitude of aerosol layers and shown in Fig. 8. The non-dust aerosol contribution is increasing the higher the aerosol layer is in the atmosphere and at the same time the fine dust contribution is decreasing, as can also be seen in Fig. 7. At this location, coarse dust contributes more at heights between 2 to 5 km, which is in line with Fig. 7 of Filioglou et al. (2020) where higher $\delta_p$ at 532 nm has been detected. An example of vertical profiles is given in Fig. 9. The fine dust and the non-dust aerosols dominant the lowest layer
below 1.5 km. The lofted layer between 3 and 4.5 km presents enhanced $\delta_p$ at both 355 and 532 nm, containing mainly coarse and fine dust. A higher disparity between $\delta_p$ at 355 and 532 nm was found for an upper layer between 5 and 6 km, revealing a mixture of coarse dust and non-dust aerosol particles.

### 3.2   Asian dust

Central Asia is one of the hot spot regions facing significant environmental challenges and climate-change effects. Hofer et al.
(2020) presented a dense data set of lidar observations for a Central Asian site during the 18 month campaign from March 2015 to August 2016, at Dushanbe, Tajikistan, in the framework of the Central Asian Dust EXperiment (CADEX) project.

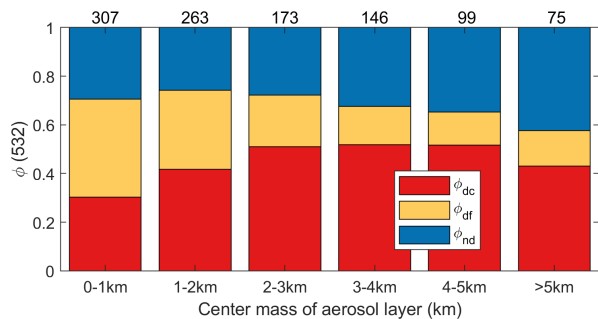

**Figure 8.** Height-dependent aerosol backscatter fractions ($\phi_x(532)$) of coarse dust (dc), fine dust (df), and non-dust (nd) aerosol, for Arabian dust layers observed over United Arab Emirates. Layer numbers are given on the top.

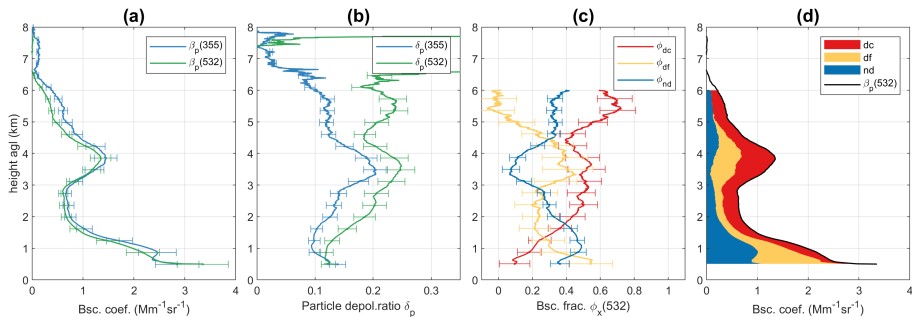

**Figure 9.** An example of lidar-derived optical profiles (time-averaged at 00:10–01:09 on 4 August 2018 over United Arab Emirates). (a) Particle backscatter coefficients ($\beta_p$) and (b) particle linear depolarization ratio ($\delta_p$) at 355 and 532 nm. (c) The retrieved backscatter fractions ($\phi_x$) of coarse dust (dc), fine dust (df), and non-dust (nd) aerosols at 532 nm, with uncertainties (shown by error bars) calculated using Monte Carlo approach following method in Sect. 2.4. (d) Separation of 3 aerosol backscatter coefficients at 532 nm.

They found broad distributions of optical properties of the 276 aerosol layers, reflecting the occurrence of very different aerosol conditions with aerosol mixtures consisting of long-range-transported dust, regional dust, and anthropogenic pollution.

Layer mean $\delta_p$ at 355 and 532 nm are shown in Fig. 10, with the layer mean $\mathring{A}_{\beta,p}$ shown by color scale, and the layer depths by marker sizes. As already discussed in Hofer et al. (2020), with decreasing extinction-related Ångström exponents, the $\delta_p$ at both wavelengths increases, which is an expected result. For the decomposition, we considered three aerosol types in those layers: coarse and fine dust (with optical properties in Table 1), and non-dust aerosols. Hofer et al. (2020) derived the mean values of $\delta_{nd}$ as $0.02 \pm 0.01$ or $0.03 \pm 0.01$ at 355 or 532 nm, respectively. $\mathring{A}_{\beta,nd}$ is assumed as 2 (cf., Table 1). The pair values of $\delta_p$ are mostly located close to the characteristic curve of coarse dust and non-dust aerosol. The aerosol backscatter fractions are calculated (Eq. 22) and shown in Fig. 11, as time series. Largest dust contributions were found during the summer season, with coarse (or fine) dust contributes 61 % (or 8 %) on the backscatter coefficients at 532 nm. During spring and autumn, dust (including coarse and fine mode) and non-dust aerosols have nearly equal contributions. Only during winter months, non-dust (e.g., urban haze) dominates with 75 % on the backscatter coefficients.

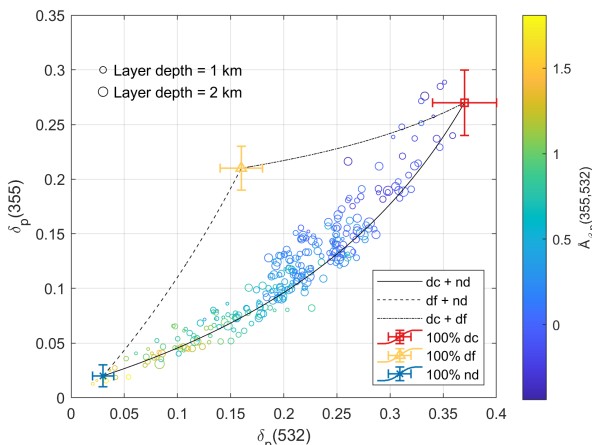

**Figure 10.** Scatter plot of layer-mean particle linear depolarization ratios ($\delta_p$) of the particle ensemble at 355 and 532 nm of Central Asian dust layers observed over Dushanbe. The layer-mean backscatter-related Ångström exponents ($\mathring{A}_{\beta,p}$) are shown by color scale, and the layer depths by marker sizes. Theoretical characteristic curves of $\delta_p$ at 355 and 532 nm for the 2 aerosol mixtures are in black. Pair values of $\delta_x$ for coarse dust (dc), fine dust (df), or spherical non-dust (nd) aerosols are shown as red square, yellow triangle, or blue cross, respectively.

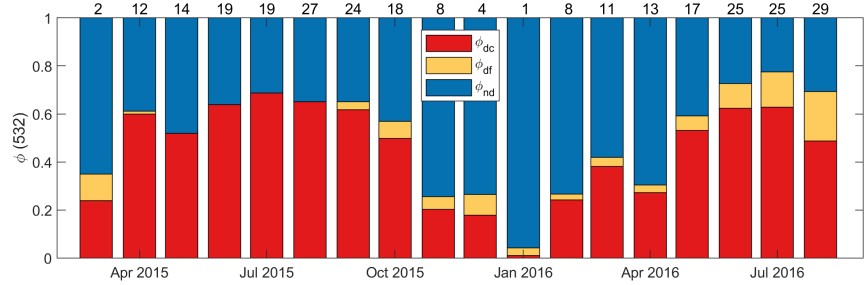

**Figure 11.** Monthly means of aerosol backscatter fractions ($\phi_x(532)$) of coarse dust (dc), fine dust (df), and non-dust (nd) aerosol, for Central Asian dust layers observed over Dushanbe. Layer numbers are given on the top.

Hu et al. (2020) documented 1 month East Asia dust aerosol observations over Kashi, China, in April 2019. The dust particles
originated mainly from the Taklamakan desert. The measured $\delta_p$ of dust layers were about $0.28$–$0.32 \pm 0.07$, $0.36 \pm 0.05$, and $0.31 \pm 0.05$ at 355, 532, and 1064 nm, respectively. These $\delta_p$ are higher than the typical values of Asian dust in the literature (Murayama et al., 2004; Dieudonné et al., 2015; Hofer et al., 2017). The reason could be linked to the fact that observations were near the dust source region, and there would be a large fraction of coarse and giant dusts particles. They described four representative cases with six dust layers. The $\delta_p$ values of two pure dust layers are 0.32 at 355 nm and 0.31 at 1064 nm, which
are higher than the particle depolarization ratios of coarse dust assumed in this paper (0.27 at 355 nm and 0.27 at 1064 nm in Table 1). For the four polluted dust layers, two of them have higher $\delta_p$ at 355 nm and 1064 nm compared to the $\delta_{dc}$. Under current assumptions, those $\delta_p$ pairs locate outside the characteristic region, which would lead to some un-physical results

(below zero or above one as the fraction). Therefore, the input parameters in Table 1 must be updated. The depolarization ratios of Taklamakan desert dust need to be determined, especially at wavelengths 355 and 1064 nm. The inclusion of giant dust particles may have impact on the depolarization ratios of coarse dust type when performing the decomposition.

### 3.3 Saharan dust

Many research studies have explored the geometric and optical properties of Saharan dust layers through lidar observations (e.g., Ansmann et al., 2003; Groß et al., 2011; Groß et al., 2015; Szczepanik et al., 2021; Gebauer et al., 2024). In this section, available lidar observations of the Saharan dust depolarization ratios at all three classical lidar wavelengths (355, 532 and 1064 nm) are considered. The small and spherical particles affect the back-scattering at shorter wavelengths more effectively than at longer wavelengths, whereas longer wavelengths are more sensitive to large particles.

Haarig et al. (2022) (referred as Ha22 further on) reported two case studies of Saharan dust layers observed over Leipzig, Germany. In the first pure-dust case in February 2021, Saharan dust plumes reached the observation station in less than 2 d after emission, exhibiting $\delta_p$ of $0.242 \pm 0.024$, $0.299 \pm 0.018$ and $0.206 \pm 0.010$ at 355, 532 and 1064 nm, respectively. In the second polluted-dust case in March 2021, the dust spent about 1 week in transportation, and the dust plume mixed with European haze. Such dust layers have $\delta_p$ of $0.174 \pm 0.041$, $0.298 \pm 0.016$ and $0.242 \pm 0.007$ at 355, 532 and 1064 nm, respectively. Haarig et al. (2017a) (referred as Ha17) documented the lidar observations in Saharan dust layers over Barbados in the summer seasons of 2013 and 2014, in the framework of the Saharan Aerosol Long-range Transport and Aerosol-Cloud-Interaction Experiment (SALTRACE). $\delta_p$ for long-range-transported (after approximately 1 week transport over the tropical Atlantic) Saharan dust layers were found to be $0.252 \pm 0.030$, $0.280 \pm 0.020$, $0.225 \pm 0.022$ at 355, 532 and 1064 nm, respectively. Hu (2018) (referred as Hu18) described two cases of lidar measurements of long-range-transported Saharan dust observed at ATOLL observatory in Lille, France. The value ranges of $\delta_p$ were 0.25–0.26 (0.25–0.27), 0.25–0.28 (0.24–0.26), 0.17–0.21 (0.20–0.22) at 355, 532, 1064 nm for the first case in March 2017 (or the second case in October 2017), respectively.

The pair values of $\delta_p$ of these cases are presented in Fig. 12 to investigate the aerosol mixing states. We assume that there are coarse dust (dc), fine dust (df), and non-dust (nd) aerosols in the mixture, with values of $\delta_x$ and $\text{Å}_{\beta,x}$ as given in Table 1. The optical properties of background non-dust aerosols for these cases are different, but here we consider them as the same for the simplification. The two cases of Ha22 are located well on the characteristic curves of dc & df, or dc & nd, respectively, for $\delta_p$ pairs both at 532 & 355 nm (Fig. 12a), and at 532 & 1064 nm (Fig. 12b). In the first case, dust plumes were directly transported towards to the station, thus, only dust (both coarse and fine mode) is contained within the observed plume. In the second case, $\delta_p$ reveal the impact of aerosol pollution mixed into the dust layers after the transportation, and it seems that the remaining dust are mainly coarse mode. The other three cases of Ha17 and Hu18 are mostly located around the characteristic curve of dc & df considering $\delta_p$ pairs at 355 & 532 nm (Fig. 12a), or located inside the enclosed characteristic region of three types regarding $\delta_p$ pairs at 532 & 1064 nm (Fig. 12b). It should be noted that the values reported by Ha17 are the average over 21 individual cases of long-range-transported dust.

Apart from the first case of Ha22, all cases were long-range-transported dust. Results reveal that coarse dust contribute a lot in these layers. The dust particle size distribution can change quickly because of the dry deposition (e.g., gravitational settling).

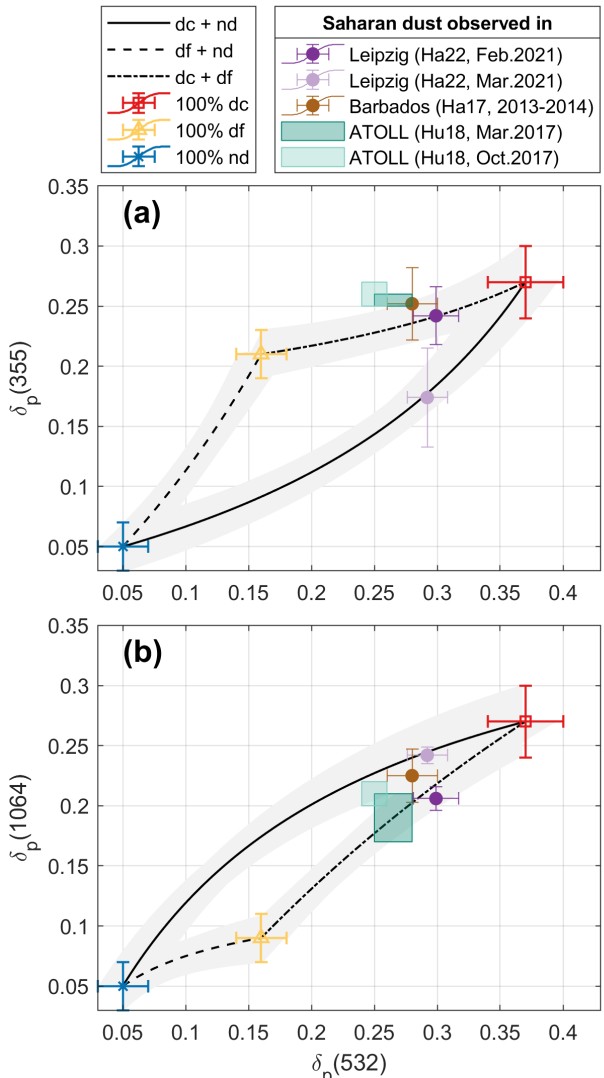

**Figure 12.** Scatter plot of layer-mean particle linear depolarization ratios ($\delta_p$) of the particle ensemble (a) at 355 and 532 nm, or (b) at 532 and 1064 nm of Saharan dust layers observed in different places: Ha22 – Haarig et al. (2022), Ha17 – Haarig et al. (2017a), Hu18 – Hu (2018). Theoretical characteristic curves of $\delta_p$ for the two aerosol mixtures are in black lines, with uncertainties shown as shaded area from Monte Carlo simulations (using parameters in Table 1). Pair values of $\delta_x$ for coarse dust (dc), fine dust (df), or spherical non-dust (nd) aerosol are shown as red square, yellow triangle, or blue cross, respectively.

However, observations have consistently shown that coarse dust, or even giant dust, are able to transport much farther than previously expected, and can have longer atmospheric lifetime (e.g., Mallios et al., 2021; Ryder et al., 2018, 2019; van der Does et al., 2018; Denjean et al., 2016).

## 4   Conclusions

Polarization-based algorithms have been developed and widely used for the decomposition of two aerosol types. In addition, some studies have extended this approach to the decomposition of three aerosol types. The measurements of depolarization ratios at multiple wavelengths from a single or multiple lidars have become increasingly common, with availability continuing to expand. Lidar networks now provide automated and standardized outputs, delivering consistent and reliable products. Among these, the particle depolarization ratio is a widely used parameter and a standard output. In this study, we present a lidar-based methodology that separates and estimates up to three aerosol-type-specific particle backscatter fractions using particle depolarization ratios measured at two wavelengths. An analytical solution is provided, resulting in an easy-to-apply algorithm. This method is applicable under two conditions: (1) the particle depolarization ratios of the aerosol types in the mixture are different, and (2) the depolarization ratios and backscatter-related Ångström exponents of each aerosol type should either be known or reasonable assumed. Therefore, laboratory and modeling studies, or field studies for a layer with only one aerosol type in atmospheric conditions, to characterize pure particles would be beneficial, and increase the accuracy for the retrievals. The relationship between particle linear depolarization ratios at two wavelengths for a mixture of two and three aerosol types has been mathematically investigated: the pair values must locate on the characteristic curved line (for two aerosol types) or remain within the enclosed region predetermined by three boundary characteristic curves (for three aerosol types). A characteristic curve line has its endpoints determined by the particle depolarization ratios of each type and its curvature influenced by the backscatter-related Ångström exponents of two aerosol types.

The present algorithm has been applied to synthetic dust mixtures and to lidar observations of Arabian dust, Asian dust, Saharan dust and their mixtures, based on the dust optical properties reported in numerous laboratory and field studies. The backscatter fractions of coarse dust (>1 $\mu$m in diameter), fine dust (<1 $\mu$m in diameter), and spherical non-dust aerosols were retrieved, which can be converted to the aerosol-type-specific backscatter coefficients, and extinction coefficients (or aerosol optical depth) with corresponding lidar ratios. These results can be further utilized to estimate the vertical profiling of mass concentration, cloud condensation nucleus (CCN) and ice-nucleating particle (INP) concentrations (Mamouri and Ansmann, 2016, 2017). Results of lidar observations from the Arabian Peninsula indicate that dust at higher altitudes tends to be coarser, whereas dust at lower altitudes is generally finer. This finding is consistent with the results of O'Sullivan et al. (2020), which report that operational models often place dust layers too low and underestimate coarse dust while overestimating fine dust. Height-resolved distributions of coarse and fine dust, retrieved using the proposed approach, can therefore serve as an important constraint for evaluating and improving model performance, offering new perspectives for a better understanding of dust transport mechanisms. It seems that for the Arabian dust and East Asia dust, the particle depolarization ratios at 355 nm for pure fine and pure coarse dust were underestimated. On the other hand, in Central Asia, the fine dust depolarization ratio at 355 nm could be lower to have more fine mode contribution. Regional differences in the pure dust depolarization ratios should be investigated in future. The uncertainty and sensitivity analyses indicate that, among type-specific optical properties, the backscatter-related Ångström exponents have only minor effects on the retrievals, whereas particle depolarization ratios have much stronger influences. Furthermore, observational uncertainties in lidar-measured particle depolarization ratios significantly

impact retrieval accuracy. Therefore, precise characterization of the lidar system is essential to ensure depolarization ratios are provided with minimal uncertainty. For the synthetic dust mixture considered in this study, uncertainties below 7 % (or 17 %) and biases below 0.01 (or 0.03) of particle depolarization ratios can be tolerated to keep the resulting fractions within uncertainties of 20 % (or 50 %).

The proposed method is well suited for investigating atmospheric aerosol mixtures containing up to three aerosol types with distinct particle depolarization ratios. Other than dust mixture, it would be also possible for the decomposition of other aerosol mixtures. For example, an aerosol mixture with spherical background particles, and two types of pollen (e.g., birch and pine pollen with different particle depolarization ratios, Shang et al., 2020; Filioglou et al., 2023). However, additional dedicated laboratory studies for the pollen characterization are desirable. Furthermore, dry marine aerosols (Haarig et al., 2017b; Ferrare et al., 2023), and stratospheric smoke aerosols (Haarig et al., 2018; Hu et al., 2019) also show enhanced particle depolarization ratios with certain spectral slopes, indicating that the algorithm is applicable to aerosol mixtures containing these types as well. This method also demonstrates strong potential for global application through existing ground-based lidar networks (e.g., EARLINET and PollyNET) and next-generation space borne lidars equipped with dual-wavelength depolarization measurements. By providing accurate height-resolved, aerosol-type-specific information on a large scale, this approach can significantly enhance regional and global aerosol characterization and support model evaluation, data assimilation, and climate studies.

*Code and data availability.* All lidar data used in this study were from previously published literature: Arabian dust (Filioglou et al., 2020), Asian dust (Hofer et al., 2020; Hu et al., 2020), and Saharan dust (Haarig et al., 2022, 2017a; Hu, 2018). These datasets are available from the corresponding authors upon request. The algorithm code is available on GitHub (https://github.com/xxshang/Shang-et-al_2026_Decomposition-of-three-aerosol-types/releases/tag/v2.0-AMT, last access: 13 January 2026) and Zenodo (Shang, 2026).

*Author contributions.* XS developed the algorithm, conceptualized the study, analyzed data, and wrote the paper. MF analyzed and provided the Arabian dust dataset under the OASIS project. JH analyzed and provided the Asian dust dataset under the CADEX project. MH, QH and PG analyzed and provided the Saharan dust dataset. SR and MK provided guidance of the study. All authors were involved in editing the paper, interpreting the results, and the discussion of the manuscript.

*Competing interests.* The authors declare that they have no conflict of interest.

*Acknowledgements.* This research has been supported by the Research Council of Finland (project no. 329216). This project has received funding from Horizon Europe programme under Grant Agreement No 101137680 via project CERTAINTY (Cloud-aERosol inTeractions & their impActs IN The earth sYstem). We acknowledge the support by the National Center of Meteorology, Abu Dhabi, UAE, under the

UAE Research Program for Rain Enhancement Science, for the Optimization of Aerosol Seeding In rain enhancement Strategies (OASIS) project. We would also like to thank Timo Anttila and Siddharth Tampi for providing on-site technical support during OASIS project. The CADEX project for first lidar measurements in Tajikistan was funded by the German Federal Ministry of Education and Research (BMBF)

in the context of "Partnerships for sustainable problem solving in emerging and developing countries" (grant no. 01DK14014). The project PoLiCyTa to setup the new lidar system in Dushanbe is also funded by BMBF under the grant number 01LK1603A. Lidar measurements in Tajikistan would not have been possible without support of the Academy of Sciences of the Republic of Tajikistan. We want to thank Sabur F. Abdullaev and Abduvosit N. Makhmudov for their support in Tajikistan, and Dietrich Althausen, Ronny Engelmann and Holger Baars for their tireless efforts to enable lidar observations in Tajikistan. The lidar observations also received funding from ACTRIS-D funded by BMBF

under grant agreements 01LK2001A and 01LK2002A under the FONA Strategy "Research for Sustainability". We acknowledge Labex CaPPA, CPER CLIMIBIO and OBS4CLIM for the financial support to lidar observations and research activities at ATOLL observatory, France. We acknowledge funding from the Leibniz Association through the Leibniz Competition for the Leibniz Junior Research Group OLALA (Optical Lab for Lidar Applications) under grant J128/2022. We thank Ulrich Theune for the MATLAB File Exchange code of Ternary plots (Ulrich Theune (2024). Ternary Plots (https://www.mathworks.com/matlabcentral/fileexchange/7210-ternary-plots), MATLAB

Central File Exchange. Retrieved October 24, 2024.). The authors sincerely thank Franco Marenco and the anonymous reviewer for their valuable input, constructive comments, and insightful suggestions, which greatly improved the quality and clarity of this paper.

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
