# Peer review of "Decomposition of three aerosol types using lidar-derived depolarization ratios at two wavelengths"

_EGUsphere, 2024_

## Referee Comment (RC2)

[revised manuscript text omitted]

**3 Case studies of mineral dust**

In this section, the algorithm was applied to lidar observations of dust at different regions. *[handwritten: As your retrieval has also a non-dust component, I would reformulate this sentence as follows: "... to lidar observations of aerosols in different dust-affected regions"]*

[revised manuscript text omitted]

↘ as also
seen from Fig. 5!

[Figure]

*Handwritten note (left margin): Å is generally associated to particle size large p. = geom. optics ⟹ Å=0 fine p. ⟹ Å=1÷2 molecules (Rayleigh sc.) ⟹ Å=4 So this result is rather expected!*

[Figure]

**Figure 8.** Scatter plot of layer-mean particle linear depolarization ratios ($\delta_p$) of the particle ensemble at 355 and 532 nm of Central Asian dust layers observed over Dushanbe. The layer-mean backscatter-related Ångström exponents ($\text{Å}_{\beta,p}$) are shown by color scale, and the layer depths by marker sizes. Theoretical characteristic curves of $\delta_p$ at 355 and 532 nm for the 2 aerosol mixtures are in black. Characteristic $\delta_x$ of coarse dust (dc), fine dust (df), or spherical non-dust (nd) aerosols are shown as red square, yellow triangle, or blue cross, respectively.

**3.2 Asian dust**

*Handwritten note: these are usually the same thing. do you mean local vs. transported dust?*

Central Asia is one of the hot spot regions facing significant environmental challenges and climate-change effects. Hofer et al.
(2020) presented a dense data set of lidar observations for a Central Asian site during the 18 month campaign from March
2015 to August 2016, at Dushanbe, Tajikistan, in the framework of the Central Asian Dust EXperiment (CADEX) project.
They found broad distributions of optical properties of the 276 aerosol layers, reflecting the occurrence of very different
aerosol conditions with aerosol mixtures consisting of mineral dust, soil dust, and anthropogenic pollution.

Layer mean $\delta_p$ at 355 and 532 nm are shown in Fig. 8, with the layer mean $\text{Å}_{\beta,p}$ shown by color scale, and the layer depths
by marker sizes. As already discussed in Hofer et al. (2020), with decreasing extinction-related Ångström exponents, the $\delta_p$
at two wavelengths increase. For the aerosol component separations, we considered three aerosol types in those layers: coarse
and fine dust (with characteristics in Table 1), and non-dust aerosols. Hofer et al. (2020) derived the mean values of $\delta_{nd}$ as 0.02
$\pm$ 0.01 or 0.03 $\pm$ 0.01 at 355 or 532 nm, respectively. $\text{Å}_{\beta,nd}$ is assumed as 2 (cf., Table 1). The pair values of $\delta_p$ are mostly
located close to the characteristic curve of coarse dust and non-dust aerosol. The aerosol backscatter fractions are calculated
(Eq. 19) and shown in Fig. 9, as time series. Highest [*longest*] dust contributions were found during the summer season, with coarse
(or fine) dust contributes 61 % (or 8 %) on the backscatter coefficients at 532 nm. During spring and autumn, dust (including
coarse and fine mode) and non-dust aerosols have nearly equal contributions. Only during winter months, non-dust (e.g., urban
haze) dominates with 75 % on the backscatter coefficients.

Hu et al. (2020) documented 1 month East Asia dust aerosol observations over Kashi, China, in April 2019. The dust particles
originate mainly from the Taklamakan desert. The measured $\delta_p$ of dust layers are [*were*] about 0.28–0.32 $\pm$ 0.07, 0.36 $\pm$ 0.05, and

*Handwritten note (bottom): "highest" could refer to higher altitude, whereas here you refer to larger contributions*

[revised manuscript text omitted]

※ use your equations to assess what max. bias and uncertainty are need for this method QUANTIFY THE REQUIRED BIAS AND UNCERTAINTY

---

## Author Comment (AC1)

**Responses to suggestions of Referee #2**

Thank you for reading the manuscript and providing very useful comments and suggestions to improve the paper. The replies to the referee comments are given below. The referee comments are in blue with our responses in black. The modifications in the revised manuscript can be found in the track change version of manuscript.

Based on the reviewers' comments, we have revised the structure of Sect. 2 and relocated some equations, sentences, and paragraphs in Sect. 2. We have added a new subsection (Sect. 2.3: Synthetic aerosol mixture), which mainly incorporates content previously included in Sects 2.1 and 2.2.

Sentences and equations that were only moved, without any changes to their wording, do not appear in the tracked changes version. This was done to ensure smoother readability.

In this manuscript, the authors propose a method for decomposing aerosol components using particle depolarization ratios measured from lidar at two different wavelengths. The methodology is presented in detail for cases involving mixtures of two and three aerosol components, respectively. Case studies are conducted for mineral dust from Arabian, Asian, and Saharan sources. The method is comprehensively described, and the case studies provide a thorough experimental validation. Given the increasing availability of multi-wavelength lidar measurements, this work represents a valuable contribution to the lidar and aerosol observation communities. However, I have one major comment regarding the methodological analysis, along with several technical and minor comments. Additionally, the manuscript would benefit from proofreading by a native English speaker to improve clarity and flow.

**Major comment:**

The proposed method, e.g. for the mixture of two components (Eqs. 5 - 9), relies on input parameters from Table 1. These parameters undoubtedly influence the results of the aerosol decomposition, and therefore, a more comprehensive sensitivity analysis that what it has for now is needed. In Section 2.3 and Table 2, the authors conduct an uncertainty analysis for three cases with different depolarization ratios at 355 and 532 nm, comparing reference results with those obtained using Monte Carlo simulations with normal distributions for all input variables. However, the analysis does not address the sensitivity of the results to individual variables. Specifically, which variables have the most significant impact on the decomposition results? For instance, the Ångström exponent is known to vary considerably from different studies, but its specific influence on the results presented in this manuscript remains unclear. While I acknowledge the authors' point that this paper primarily aims to present a method rather than investigate aerosol characteristics (which is beyond the scope of this work), understanding the sensitivity of the results to the input parameters is crucial. Such an analysis would help identify which parameters require more careful consideration in future applications. Furthermore, a sensitivity study could provide a statistical explanation for why certain points in Figure 5 deviate from the curve.

The reviewer raises a valid point. Indeed, the particle depolarization ratios (PDRs) at two wavelengths as well as the relative backscatter-related Ångström exponent (BAE) defines the observational space required by the proposed decomposition method. Therefore, knowledge of the parameters in Table 1 is of paramount importance for maximizing the confidence of the aerosol backscatter fractions in the two- (or three-) type aerosol mixture.

We have performed a more comprehensive sensitivity analysis, and revised the Sect. 2.4 Uncertainty study. In the revised manuscript, we present the sensitivity analysis in 4 parts:

1. A global sensitivity analysis to assess the combined effect of all input variables. Similar as previous version, but adding uncertainties on $\delta p$ as well, thus considering 11 variables in the revised version.
2. Individual sensitivity analysis using the one-at-time (OAT) method, to assess the influence of each variable independently.
3. We change the uncertainty levels on each variable to investigate the influences.
4. We performed an additional analysis on observational parameters $\delta p$, to study their tolerated bias.

From the individual sensitivity analysis, we found that the BAEs have only minor effects on the retrieval compared to other parameters. This finding is important given the considerable variability commonly observed in Ångström exponents. Such results are also added in the conclusion.

Apart from the revised sensitivity analysis present in the revised manuscript, we have performed additional analysis to investigate the effect of PDRs and BAEs.

In Fig. 1d we observe that the characteristic PDRs define the observational space in which the decomposition method is applied while the BAE affects the curvature between these edge points. Equivalent to Fig. 1d, the figure below shows the sensitivity of the BAE for an aerosol mixture of three aerosol types. All possible BAE combinations between -1 and 1, 0 and 2, and 1.5 and 2.5 with an increment of 0.1 for each BAE were considered for coarse, fine, non-dust types, respectively.

[Figure]

Similar to Figure 1d featuring three aerosol components.

To visualize further the effects, we have made a sensitivity study where one group of parameters is changed at a time from the ones found in Table 1. In Table 1 there are two groups, one that includes all the PDRs and a second one for all the BAEs. Therefore, we have performed two simulations. In each simulation, one group of parameters (e.g., the PDRs) was varied while the second group (e.g., the BAE) was kept at the fixed mean value as marked in Table 1 without accounting for its uncertainty.

Here we assume a 30% uncertainty on PDRs or BAEs as an example. The first simulation includes 77 different combinations where the PDR of coarse, fine and non-dust types was between 0.24 and 0.30 (0.34 and 0.39), 0.19 and 0.23 (0.14 and 0.19), and 0.03 and 0.07 (0.03 and 0.07) with 0.01 increment at 355 nm (532 nm) wavelength, and the fixed BAE coarse, fine and non-dust was -0.2, 1.5 and 2.0 between 355 and 532 nm, respectively. For the second simulation the BAE ranges at $-0.2 \pm 0.06$, $1.5 \pm 0.45$ and $2.0 \pm 0.6$ for the coarse, fine and non-dust types, respectively.

For physical meaningfulness, we applied the two simulations to case 1 and case 2 without accounting for the uncertainty of the measured PDR. Cases 1 and 2 are located inside the observational space but at different distances from the edges. Visualizing the effect, we observe that varying the edges of the observational space (PDRs) induces higher uncertainty (wider distributions) than varying the BAEs for the same 30% uncertainty. Therefore, accurate knowledge of the parameters in Table 1 as well as adequate assumption of the aerosol mixture in the atmosphere is important. At the same time, the uncertainty of the measure PDR should be kept within acceptable levels.

[Figure]

Estimated uncertainties varying (top) PDRs or (bottom) BAEs of each aerosol type for cases 1 and 2.

Regarding the points deviating from Fig. 5, we would like to emphasize that different atmospheric studies often use different definitions for the geometrical boundaries of the aerosol layer. These differences, combined with the vertical smoothing applied to optical profiles, can influence the range of particle depolarization ratios reported. Most commonly, the optical properties are reported as mean values with a standard deviation. In Filioglou et al. (2020), the maximum particle depolarization ratios reported were 32% and 35% at 355 and 532 nm, respectively (see Table 1) while mean values amounted to $25 \pm 2$ and $31 \pm 2$, respectively. Then, recent laboratory studies focused on the chemical composition of dust particles and their optical effect reporting particle depolarization ratios at 355nm up to 32% depending on the dust mixture (Miffre et al. 2023). A thorough conversation of the selection of the optical properties in Table 1 is already included in the manuscript, and it is out of the scope of this manuscript to define the characteristics of fine and coarse aerosol particles. This manuscript serves as a demonstration of the methodology, and provide an easy-to-apply algorithm. The optical properties of individual aerosol type can be readily updated as more accurate values become available and new observations emerge in the field.

Filioglou, M., Giannakaki, E., Backman, J., Kesti, J., Hirsikko, A., Engelmann, R., O'Connor, E., Leskinen, J. T. T., Shang, X., Korhonen, H., Lihavainen, H., Romakkaniemi, S., and Komppula, M.: Optical and geometrical aerosol particle properties over the United Arab Emirates, Atmos. Chem. Phys., 20, 8909–8922, https://doi.org/10.5194/acp-20-8909-2020, 2020.

Miffre, A., Cholleton, D., Noël, C., and Rairoux, P.: Investigating the dependence of mineral dust depolarization on complex refractive index and size with a laboratory polarimeter at 180.0° lidar backscattering angle, Atmos. Meas. Tech., 16, 403–417, https://doi.org/10.5194/amt-16-403-2023, 2023.

**Other comments:**

The abstract of this manuscript needs to be improved. Lines 4 - 6: The authors state the advantages of the proposed method, but the logic may confuse readers. Specifically, "And it requires the proper knowledge of characteristic depolarization ratio and the backscatter-related Ångström exponent of each aerosol type" is a prerequisite for the method, not an advantage. Please rephrase these sentences to clarify the distinction between prerequisites and advantages.

Thank you for your comment. We have rewritten the abstract as below:

"Lidar-based algorithms for aerosol-type separation have the potential to improve air-quality assessments, estimates of aerosol direct and indirect radiative forcing, and the detailed characterization of their vertical distribution. In this study, we present an easy-to-apply algorithm that employs lidar-derived particle linear depolarization ratios measured at two wavelengths to separate up to three aerosol-type-specific particle backscatter fractions. These fractions are estimated under the assumptions that the depolarization ratios of each aerosol type in the mixture differ, and that both the depolarization ratios and the backscatter-related Ångström exponents at two wavelengths for each aerosol type are known. The mathematical relationship between particle linear depolarization ratios at two wavelengths for an aerosol mixture has been derived and expressed as a system of equations. These equations define the region of the observational space that can be meaningfully populated, with boundaries determined by the depolarization ratios and backscatter-related Ångström exponents of the pure aerosol types. Data collected in the Arabian Peninsula confirmed the predicted region of the observational space. The proposed algorithm is applied to synthetic dust mixtures as well as to atmospheric lidar observations of Arabian dust, Asian dust, Saharan dust and their mixtures, with the goal of decomposing coarse-mode dust, fine-mode dust, and low-depolarizing non-dust aerosols. We also discuss the impact of uncertainties in the prior optical properties of the pure aerosol types, along with the effects of observational uncertainties and biases. Overall, the method enhances the potential of dual-wavelength depolarization measurements for improving our understanding of the vertical distribution of coarse and fine dust"

Line 12: The claim that the method is "more accurate than the common use of the ratio of the particle linear depolarization ratios" requires statistical support. Please provide evidence or references to substantiate this statement.

Thank you for your comment.

In the literature, the ratio of particle linear depolarization ratios (PDRs) at different wavelengths is often used, and often referred as a constant value. In this study, we derived the mathematical relationship between PDRs at two wavelengths and demonstrated that this relationship is not linear. Hence, the ratio is not constant, but instead follows a curved dependence.

We have reformulated the abstract, the specific sentence in the abstract is now removed.

In Sect. 2.1, we have modified the text as:

"It is concluded that the relationship between δp at two wavelengths is not linear. Thus, the commonly used ratio of δp(λ1) and δp(λ2), which is typically assumed to be constant, is less accurate than the characteristic curved relationship proposed in this study."

Line 35: The sentence is unclear and should be rephrased for better readability.

The sentence has been rephrased as suggested.

"The POLIPHON method has been applied to many lidar observations across the world, benefiting from the wide availability of these single-wavelength polarization lidars."

Line 63: The description of Eqs. 1-2 is confusing. The statement, "the calculation involves the aerosol backscatter coefficient (βx) and aerosol-type-specific characteristic depolarization ratio (δx)," implies that βx is not aerosol-type-specific, but I understand βx is also aerosol-type-specific.

Thank you for your comment.

Initially, we intended to state that δx is specific to the aerosol type, while βx depends on the amount of that aerosol type present in the mixture and therefore does not have a fixed value (or characteristic value).

We have modified the sentence for improved clarity.

"In this context, each particle consists of a single aerosol type, and the calculation involves the aerosol backscatter coefficient (βx) and the particle depolarization ratio (δx), where the index x corresponds to each aerosol type (a, or b, or c), or to the mixture (p)."

Line 74: The "Methodology" section requires improvement. The authors list several equations (Eqs. 1-4) but do not systematically introduce the proposed method or explain how these parameters are used to develop the algorithm. The statement, "To apply the novel algorithm for the decomposition of two or three aerosol components," is premature, as the algorithm has not yet been clearly defined. Readers must read the entire manuscript to understand how the parameters are used for aerosol separation. The authors should explicitly introduce the algorithm before discussing its application.

Thank you for your suggestion.

We have revised the structure of Sect. 2 and relocated some paragraphs and sentences in Sect. 2. We have added a new subsection (Sect. 2.3: Synthetic aerosol mixture), which mainly incorporates content previously included in Sects 2.1 and 2.2.

At the beginning of the methodology section, we have added the following introductory paragraph:

"The basic concept is to use lidar-derived particle linear depolarization ratios obtained from a multi-wavelength lidar (or two separate instruments), together with two key optical properties (namely, the particle depolarization ratios and the backscatter-related Ångström exponents for pure aerosol types), to separate the aerosol mixture into its individual aerosol types. In this section, we introduce the algorithm and the corresponding set of equations for decomposing mixtures of two or three aerosol types (Sects. 2.1-2.2). The algorithm is then applied to synthetic aerosol mixtures in Sect. 2.3, followed by a comprehensive sensitivity and uncertainty analysis in Sect. 2.4."

We hope that these modifications have made the methodology section clearer to the reader.

Line 119: The statement, "Assuming the same lidar ratios at 355 and 532 nm, these values can be used for the Åβ(355,532)," requires clarification. Please briefly introduce the lidar ratio, and also provide references supporting the assumption that lidar ratios are the same (or very close) at 355 nm and 532 nm.

Thank you for your comment. We added the following text to the manuscript:

"The lidar ratio, defined as the extinction-to-backscatter ratio, has been widely used in lidar-based aerosol classification algorithms because it provides information on aerosol type. Numerous lidar studies have investigated the spectral dependence of the lidar ratio for different aerosol types (e.g., Haarig et al., 2025). For instance, Floutsi et al., (2023) present a comprehensive collection of depolarization ratios, lidar ratios, and Ångström exponents for various aerosol types and mixtures based on ground-based lidar observations. For most aerosol types, including dust from most regions except Central Asia, the assumption of lidar ratio equality between 355 and 532 nm is generally valid within observational uncertainties. However, for smoke mixtures, this assumption should be applied with caution."

Floutsi, A. A., Baars, H., Engelmann, R., Althausen, D., Ansmann, A., Bohlmann, S., Heese, B., Hofer, J., Kanitz, T., Haarig, M., Ohneiser, K., Radenz, M., Seifert, P., Skupin, A., Yin, Z., Abdullaev, S. F., Komppula, M., Filioglou, M., Giannakaki, E., Stachlewska, I. S., Janicka, L., Bortoli, D., Marinou, E., Amiridis, V., Gialitaki, A., Mamouri, R.-E., Barja, B., and Wandinger, U.: DeLiAn – a growing collection of depolarization ratio, lidar ratio and Ångström exponent for different aerosol types and mixtures from ground-based lidar observations, Atmos. Meas. Tech., 16, 2353–2379, https://doi.org/10.5194/amt-16-2353-2023, 2023.

Haarig, M., Engelmann, R., Baars, H., Gast, B., Althausen, D., and Ansmann, A.: Discussion of the spectral slope of the lidar ratio between 355 and 1064 nm from multiwavelength Raman lidar observations, Atmos. Chem. Phys., 25, 7741–7763, https://doi.org/10.5194/acp-25-7741-2025, 2025.

**Minor comments:**

Line 38: Specify "the 532 nm and 355 nm wavelengths" by adding, for example, "of lidar instruments."

Thank you for your suggestion. We have rephrased the sentence to:

"Most commonly, the 532 nm and 355 nm wavelengths have been used to perform lidar-derived depolarization ratio measurements."

Line 46: Replace "at 532 or 355 nm" with "at 532 and 355 nm".

Replaced according to reviewer's suggestion.

Line 207: "idea" -> "ideal"?

Corrected according to reviewer's suggestion.

---

## Author Comment (AC2)

**Responses to suggestions of Franco Marenco**

Dear Franco Marenco,

Thank you for reading the manuscript and providing very useful comments and suggestions to improve the paper. The replies to your referee comments are given below. The referee comments are in blue with our responses in black. The modifications in the revised manuscript can be found in the track change version of manuscript.

Based on the reviewers' comments, we have revised the structure of Sect. 2 and relocated some equations, sentences, and paragraphs in Sect. 2. We have added a new subsection (Sect. 2.3: Synthetic aerosol mixture), which mainly incorporates content previously included in Sects 2.1 and 2.2.

Sentences and equations that were only moved, without any changes to their wording, do not appear in the tracked changes version. This was done to ensure smoother readability.

The article by Xiaoxia Shang and co-authors describes a new algorithm that allows combining several optical parameters observed by lidar at two wavelengths, to decompose an external mixture of three aerosol types, quantitavely. This new method builds on pre-existing methodologies using a single wavelength to separately quantify two aerosol types. The method assumes prior knowledge of a number of intensive optical properties: the depolarisation ratio of the three pure aerosols at each wavelength, as well as their backscatter Angstrom exponent (Table 1). A system of linear and quadratic equations is derived, which can be resolved in terms of the backscatter fraction of each of the three aerosol components. The equations define the region of the observational space that can be meaningfully populated, and the observational data do confirm the accuracy of this prediction (Figure 5). The method is successfully demonstrated for a number of aerosol mixtures, observed with differing lidar systems, in the context of dust aerosols from the Arabian Peninsula, Asia and the Sahara, mixed with other aerosols. A succinct error analysis is also included.

I definitely think that the method proposed is promising and that this article is worth publishing. I however also feel that the paper can be substantially improved with a little more in-depth analysis of some points: better mathematical analysis of the system of equations, better quantification of the observational bias and uncertainty requirements, better use of the data collected for an update of the assumptions of Table 1, better highlight of a side scientific result as explained below, and stronger conclusions. The abstract needs moreover more work to make it self-understandable. With a little additional work, I believe that the paper will become a highly-cited reference paper for a wide number of applications.

MAJOR POINTS:

1. The second half of the abstract is a little hard to follow. My suggestion is as follows (from line 6): "The mathematical relationship between particle linear depolarization ratios at two wavelengths for a mixture of aerosol components has been derived and expressed as a system of equations. The equations define the region of the observational space that can be meaningfully populated and its boundaries: the latter are determined by the depolarization ratios of the pure aerosol components, and their backscatter-related Ångström exponents. Data collected in the Arabian Peninsula confirmed the predicted region of the observational space, and resolving the system of equations allowed us to quantify the contribution of each aerosol component. The novel algorithm has been applied to synthetic dust mixtures and to actual lidar observations of Arabian dust, Asian dust, and Saharan dust, so as to decompose coarse-mode dust, fine-mode dust, and low-depolarising non-dust. The impact of uncertainties in the prior optical properties of the pure-aerosol components are discussed, together with impact of observational uncertainties and biases. The method that we propose offers a promising role of dual-wavelength depolarisation measurements for the understanding the vertical distribution of fine and coarse dust."

Thank you for your comment and suggestion. We have modified the abstract as follows:

"Lidar-based algorithms for aerosol-type separation have the potential to improve air-quality assessments, estimates of aerosol direct and indirect radiative forcing, and the detailed characterization of their vertical

distribution. In this study, we present an easy-to-apply algorithm that employs lidar-derived particle linear depolarization ratios measured at two wavelengths to separate up to three aerosol-type-specific particle backscatter fractions. These fractions are estimated under the assumptions that the depolarization ratios of each aerosol type in the mixture differ, and that both the depolarization ratios and the backscatter-related Ångström exponents at two wavelengths for each aerosol type are known. The mathematical relationship between particle linear depolarization ratios at two wavelengths for an aerosol mixture has been derived and expressed as a system of equations. These equations define the region of the observational space that can be meaningfully populated, with boundaries determined by the depolarization ratios and backscatter-related Ångström exponents of the pure aerosol types. Data collected in the Arabian Peninsula confirmed the predicted region of the observational space. The proposed algorithm is applied to synthetic dust mixtures as well as to atmospheric lidar observations of Arabian dust, Asian dust, Saharan dust and their mixtures, with the goal of decomposing coarse-mode dust, fine-mode dust, and low-depolarizing non-dust aerosols. We also discuss the impact of uncertainties in the prior optical properties of the pure aerosol types, along with the effects of observational uncertainties and biases. Overall, the method enhances the potential of dual-wavelength depolarization measurements for improving our understanding of the vertical distribution of coarse and fine dust."

2. I have not done a full analysis of the system of equations proposed by the authors, but I am not persuaded by their statement that there is always a unique solution (lines 163 and 184). What I see is a system of 6 equations and 4 unknowns (two aerosols) or 7 equations and 6 unknowns (three aerosols). One equation (sum of phis for lambda1) has been omitted and should also be included. Moreover, equation 10 is not independent of equations 5-9, so I don't see how it can reduce the number of constrains. In other words, I think that a full mathematical analysis of the existence and unicity of the solution should be added in the paper. It is possible that the statement that "there is always a unique solution" will be found to be correct, but in the current version this is not explained or demonstrated.

Thank you for the comment, we have revised Sect. 2 to provide a clearer and more comprehensive explanation of the methodology.

For a mixture of two aerosol types, two equations at a single wavelength are sufficient for the decomposition (2 eqs for 2 unknowns), which has been used in many studies. We present this approach in an alternative form, via a system of equations at two wavelengths, specifically to derive the mathematical relationship between lidar-derived particle depolarization ratios at two wavelengths. We have revised the text to make this point clearer in the revised version.

For a mixture of three aerosol types, a system of 7 equations is used (eqs.14-20 in the revised version). Among them, there are only 6 independent equations, because any one of eqs.17-20 can be derived from the other three (4 equations, but only 3 independent ones). This point has been clarified in the revised version.

3. The text on the observational uncertainties (lines 223-231) can definitely benefit from an expanded discussion aiming at quantifying observational requirements. Readers will need to know what observational accuracy they need to achieve so as to be able to apply the method proposed by the authors. I would start from what are the average and best measurement uncertainties on delta_p in the literature and finding how they impact the retrieval of the three components. I would then suggest to exploit the equations developed in the article to assess and quantify what are the maximum bias and uncertainty that can be tolerated for depolarisation measurements, when applying the method (this could be more useful than the generic statement that a "small uncertainty" is necessary (line 331).

Thank you for the comment.

We have conducted a more comprehensive sensitivity analysis, and revised the Sect. 2.4 Uncertainty study. In the revised manuscript, we present the sensitivity analysis in 4 parts:

1. A global sensitivity analysis to assess the combined effect of all input variables. Similar as previous version, but adding uncertainties on $\delta p$ as well, thus considering 11 variables in the revised version.
2. Individual sensitivity analysis using the one-at-time (OAT) method, to assess the influence of each variable independently.

3.  We change the uncertainty levels on each variable to investigate the influences.
4.  We performed an additional analysis on observational parameters δp, to study their tolerated bias.

Descriptions about the updated sensitivity analysis on observational uncertainties, as well as discussions on the bias and uncertainty for the depolarization measurements have been added in Sect. 2.4. The results of the sensitivity study depend on the initial values of the measurable particle linear depolarization ratios (PDRs), and vary from case to case. Thus, we present the sensitivity study based on selected cases. We have also added such observational requirements in the conclusion.

4.  When the authors encounter conditions that challenge the assumptions from Table 1, such as on lines 245-246 or line 282, I suggest to try sing the values suggested by the data themselves as a-priori, to see if there can be an improvement. I would not agree that the algorithm becomes unsuitable (line 282): it is more a case of improving the inputs, and the data collected do contain the information to do this. Also, the generic statement about the effect of averaging (lines 246-248) could be substantiated with verifying how things really are with and without the averaging (either using the original measured data, or if these are not available anymore, with a simulation).

Thank you for the suggestion. We have modified the text for clarity, and added an experimental region in Fig. 7 as well as the descriptions:

"Alternatively, the measurement data can serve as a prior information for determining particle depolarization ratios of each aerosol type. For instance, to include as many measured pairs as possible inside the characteristic region, we can assume $\delta_{dc}(355)$ and $\delta_{df}(355)$ as 0.31 and 0.23, respectively. At the same time, the curvature of the curves needs to be increased by raising $A_{\beta,nd}(355, 532)$ to 2.5. The resulting characteristic region is indicated by gray dotted lines in Fig. 7. This adjustment appears to yield an improvement; however, it represents only a preliminary attempt and was not adopted in the subsequent analyses."

[Figure]

For line 282, we have changed text for clarity:

"Under current assumptions, those δp pairs locate outside the characteristic region, which would lead to some un-physical results (below zero or above one as the fraction). Therefore, the input parameters in Table 1 must be updated."

In the discussion about the effect of averaging (lines 246-248), we refer to the values reported in the literature, which were used to derive the input parameters presented in Table 1.

We made the correction for clarity:

"It may also be that the commonly reported pure values in the literature, based on field measurements, are often layer-mean values derived from smoothed optical parameters, which may underestimate the true

characteristic particle depolarization ratios of dust particles. These values were used as inputs for Table 1 in our calculations."

5. Figure 5 shows a very interesting scientific result which it may be worth discussing in the article better, although it isn't the main focus of the article itself: for this dataset, the figure shows that dust with a higher centre of mass altitude is coarse (mixed with non-dust) and dust with a lower centre of mass is fine. True that something along these lines is later discussed with Figure 6, but I would say that it is Figure 5 that shows this result at best and it is worth highlighting (and repeat in the conclusions).

Thank you for the comment. We have added discussion:

"At higher altitudes, the $\delta_p$ pairs tend to lie closer to the characteristic curve of dc & nd, whereas layers at lower altitudes are located nearer to the characteristic curve of df & nd. This pattern suggests that dust with a higher altitude is generally coarser (mixed with non-dust particles), while dust with a lower altitude is finer."

And in the conclusion, we have added:

"Results of lidar observations from the Arabian Peninsula indicate that dust at higher altitudes tends to be coarser, whereas dust at lower altitudes is generally finer. This finding is consistent with the results of O'Sullivan et al., (2020), which report that operational models often place dust layers too low and underestimate coarse mode dust while overestimating fine mode dust. Height-resolved distributions of coarse and fine dust, retrieved using the proposed approach, can therefore serve as an important constraint for evaluating and improving model performance, offering new perspectives for a better understanding of dust transport mechanisms."

6. The conclusions as formulated are rather weak: they are mainly a summary of the article with only the last 10 lines discussing some connections with the existing literature. These conclusions should be strengthened: I believe that the method proposed could have a sensible impact on future research, therefore this could be discussed. Some points to consider for discussion in the conclusions:

   o Potential atmospheric situations where the method could bring an advantage (this is in part attempted in the current version)

   o Scientific questions that can be addressed (for example the height-distribution of coarse and fine dust as in Figure 5: this has in turn an application on e.g. model evaluation – see O'Sullivan et al, https://doi.org/10.5194/acp-20-12955-2020 – and can bring whole new perspectives on understanding better the mechanisms of dust transport)

   o Observational requirements (accuracy and bias of the lidar observations of depolarisation) to be discussed in the framework of the state of the art

   o Potential application of the method on a global scale for existing lidar networks and satellite missions

Thank you for your comments and suggestions. We have revised and strengthened the conclusion.

MINOR PONTS:

See attached annotated manuscript which contains a number of suggestions and corrections.

Thank you very much for your detailed suggestions and corrections, we have revised the manuscript accordingly.

Kind regards,

Franco Marenco

---

## Author Response (AR2)

**Summary of Minor Changes in Final Manuscript**

During the final preparation of files, we made the following minor changes:

1. Line 88: Removed the extra closing parenthesis ")".

2. Code and data availability: Updated text to include lidar data sources and added Zenodo reference for the archived code.

3. Acknowledgements: Changed "Academy of Finland (projects nos. 329216)" to "Research Council of Finland (project no. 329216)" to reflect the updated official English name.

4. References: Corrected Shimizu et al., 2004 and added a new reference for the Zenodo code archive (Shang et al. 2026).

No changes were made to equations, figures, tables, or scientific content.

The ROR "Finnish Meteorological Institute (Helsinki, Finland)" is correct.

It refers to the official institution, which is headquartered in Helsinki. Our research group is part of the same institute but located at the Atmospheric Research Centre of Eastern Finland in Kuopio.